# An Injectable Hyaluronan–Methylcellulose (HAMC) Hydrogel Combined with Wharton’s Jelly-Derived Mesenchymal Stromal Cells (WJ-MSCs) Promotes Degenerative Disc Repair

**DOI:** 10.3390/ijms21197391

**Published:** 2020-10-07

**Authors:** Un Yong Choi, Hari Prasad Joshi, Samantha Payne, Kyoung Tae Kim, Jae Won Kyung, Hyemin Choi, Michael J. Cooke, Su Yeon Kwon, Eun Ji Roh, Seil Sohn, Molly S. Shoichet, Inbo Han

**Affiliations:** 1Department of Neurosurgery, CHA Bundang Medical Center, CHA University School of medicine, Seongnam-si 13496, Korea; nschoiuy@gmail.com (U.Y.C.); hariprasadjoshi10@gmail.com (H.P.J.); kyungjaewon88@gmail.com (J.W.K.); littlechoi88@gmail.com (H.C.); syunkwon@naver.com (S.Y.K.); morolro@naver.com (E.J.R.); sisohn@cha.ac.kr (S.S.); 2Department of Biomedical Engineering, Tufts University, Medford, MA 02155, USA; Samantha.payne@tufts.edu; 3Department of Neurosurgery, School of Medicine, Kyungpook National University, Daegu 41944, Korea; nskimkt7@gmail.com; 4Department of Neurosurgery, Kyungpook National University Hospital, Daegu 41404, Korea; 5Department of Chemical Engineering and Applied Chemistry, University of Toronto, 200 College Street, Toronto, ON M5S 3E5, Canada; mikejcooke@amacathera.ca (M.J.C.); molly.shoichet@utoronto.ca (M.S.S.); 6Donnelly Centre, University of Toronto, 160 College Street, Toronto, ON M5S 3E1, Canada

**Keywords:** mesenchymal stromal cell, Wharton jelly, hyaluronic acid, methylcellulose, intervertebral disc degeneration, regeneration, extracellular matrix

## Abstract

Intervertebral disc (IVD) degeneration is one of the predominant causes of chronic low back pain (LBP), which is a leading cause of disability worldwide. Despite substantial progress in cell therapy for the treatment of IVD degeneration, significant challenges remain for clinical application. Here, we investigated the effectiveness of hyaluronan–methylcellulose (HAMC) hydrogels loaded with Wharton’s Jelly-derived mesenchymal stromal cell (WJ-MSCs) in vitro and in a rat coccygeal IVD degeneration model. Following induction of injury-induced IVD degeneration, female Sprague-Dawley rats were randomized into four groups to undergo a single intradiscal injection of the following: (1) phosphate buffered saline (PBS) vehicle, (2) HAMC, (3) WJ-MSCs (2 × 10^4^ cells), and (4) WJ-MSCs-loaded HAMC (WJ-MSCs/HAMC) (*n* = 10/each group). Coccygeal discs were removed following sacrifice 6 weeks after implantation for radiologic and histologic analysis. We confirmed previous findings that encapsulation in HAMC increases the viability of WJ-MSCs for disc repair. The HAMC gel maintained significant cell viability in vitro. In addition, combined implantation of WJ-MSCs and HAMC significantly promoted degenerative disc repair compared to WJ-MSCs alone, presumably by improving nucleus pulposus cells viability and decreasing extracellular matrix degradation. Our results suggest that WJ-MSCs-loaded HAMC promotes IVD repair more effectively than cell injection alone and supports the potential clinical use of HAMC for cell delivery to arrest IVD degeneration or to promote IVD regeneration.

## 1. Introduction

Chronic low back pain (LBP) is the leading cause of disability worldwide and the social and economic impact is enormous [1]. Not all degenerated discs exhibit chronic LBP, however, intravertebral disc (IVD) degeneration is considered as a major cause of chronic LBP [1,2,3,4]. IVD degeneration is a complex and multifactorial process, influenced by genetic, nutritional, and mechanical factors [5]. As the stage of degeneration progresses, the production of pro-inflammatory molecules, including tumor necrosis factor (TNF)-α, interferon gamma (IFN-γ), and interleukin (IL-1β, IL-6, IL-1α and IL-2) increases [5]. These molecules can be produced by both IVD cells and immune cells, such as macrophages [6], and are known to be associated with discogenic back pain [6]. IVD degeneration is characterized by the loss of IVD cells and extracellular matrix (ECM) such as aggrecan and collagen type II, with the upregulation of matrix metalloproteinases (MMPs) and inflammatory mediators leading to progressive and irreversible damage of IVD structure [1,3,4,6,7,8]. Eventually, IVD degeneration can lead to the onset of additional spinal conditions, including disc herniation, spinal stenosis, facet joint osteoarthritis, and spondylolisthesis [1,3,9,10]. Current conservative and surgical treatments focus on symptomatic relief rather than arresting degeneration or restoring the structure and function of the degenerated disc [3,11]. Furthermore, the effect of surgical treatments may be temporary, and recurrent or persistent LBP can develop [5]. Therefore, there is a strong clinical demand for the development of regenerative therapy to repair degenerated disc or arrest IVD degeneration at earlier stages [1,4]. Among the biological approaches for disc repair, cell therapy has gained interest because it offers disc regeneration potential [12,13,14]. Especially, mesenchymal stromal cells (MSCs) have been considered as a potentially ideal cell source for IVD regeneration as MSCs display broad immunomodulatory properties and the ability to differentiate into cartilage [5]. Several clinical studies on IVD cell therapies reported that MSC implantation can regenerate the degenerated disc and cure discogenic back pain [5,12].

Despite some positive outcome of MSC-based therapies, the strength of the evidence for use of MSCs in IVD degeneration is low, due to the significant risk of bias, small sample size, and lack of a control group in clinical trials [5]. Furthermore, for the successful development of MSC-based cell therapy for disc repair, there are still many obstacles to overcome such as the inflammatory milieu of the degenerated disc. For successful regeneration of the IVD, the selection of a proper cell type and source is crucial. Wharton’s jelly-derived MSCs (WJ-MSCs) have gained significant interest and attention for clinical application because of their hypoimmunogenicity and immunomodulatory potential compared with MSCs from other sources [4,15,16,17,18]. WJ-MSCs exhibit very low expression of human leukocyte antigen (HLA) class I and do not express class II (HLA-DR) as well as co-stimulatory antigens such as CD86, CD80, the key players to activate B cell and T cell [19,20]. Furthermore, WJ-MSCs synthesize extensive amounts of tolerogenic IL-10, up-regulate TGF-β and express HLA-G [21]. Their low cellular immunogenicity is advantageous for allogeneic and xenogeneic transplantation. WJ-MSCs also possess immunomodulatory potential [18,22]. Their immunomodulatory mechanism can be explained as: upregulation of negative co-stimulatory ligands; secretion of immunosuppressive soluble factors; generation of memory cells; cell fusion to escape recognition; immune avoidance mechanisms specific to fetal-maternal interface; attenuation of antigen-presenting cell functions; altered migration of immune cells; T cell energy apoptosis tolerance [18,22]. Thus, WJ-MSCs have been considered to be optimal candidates for cellular therapies in allogenic transplantation and used for treatment of various diseases (e.g., cancer, chronic liver disease, cardiovascular diseases, cartilage and tendon injury, immune mediated disorders, such graft versus host disease and systemic lupus erythematosus) [18,22].

Another strategy for successful IVD regeneration, biomaterial technology has gained increasing attention as a vehicle for cell delivery to promote the viability and differentiation of MSCs by providing a three-dimensional (3D) microenvironment and to prevent cell leakage and reduce the risk of osteophyte formation [4,12,14]. In particular, an injectable, biocompatible, and biodegradable hydrogel scaffold composed of hyaluronan (HA) and methylcellulose (MC) (HAMC) is of particular interest for therapeutic cell delivery. The HAMC hydrogel is biodegradable, water soluble and provides a three-dimensional (3D) microenvironment for the cells, thereby enhancing cell viability (Figure 1A) [23,24]. Over the last decade, Shoichet lab has developed HAMC for various neurological disease models, such as ischemic stroke [24], retinal degeneration [25] and spinal cord injury [23,26]. HAMC spatially localizes the drug or cells of interest at the site of delivery and promotes short-term controlled release of drugs to the central nervous system (CNS), with a degradation time of approximately 3 to 7 days in vivo [27,28,29,30]. Importantly, it shows low immunoreactivity upon delivery and due to the property of inverse thermal gelling, cells can be mixed into HAMC at room temperature and when delivered it will increase in viscosity to retain cells at the site of injection [31].

In our previous study, we first confirmed that biomaterial loaded with human WJ-MSCs has good regenerative potential for IVD repair in a rabbit model of disc degeneration [4]. WJ-MSCs were chosen because of several advantages for clinical use, such as robust immunomodulatory properties [4,15,16]. Additionally, we chose TissueFill (cross-linked HA derivatives; CHA Meditech Co., Ltd., Dajeon, Korea), a commercially available dermal filler as a cell carrier because there was no commercialized biomaterial for IVD repair. In the present study, we have investigated whether WJ-MSCs-loaded HAMC could induce IVD repair more effectively than cell alone injections and support the potential clinical use of HAMC for cell delivery for IVD repair.

## 2. Results

### 2.1. Quality Control of WJ-MSCs

The WJ-MSCs in the second, fourth, and seventh generations showed a spindle fibroblast-like shape (Appendix A). After induction with adipogenic medium, WJ-MSCs gradually changed from fibroblast-like cells to flattened cells, and lipid droplets accumulated within them. The adipogenic differentiated MSCs were visualized by staining with Oil red-O on day 15. After incubation with osteogenic or chondrogenic medium for 15 days, MSCs were positive for alizarin red and alcian blue staining, respectively (Appendix A). The cell population doubling time was calculated three times. The cell doubling time in the second, fourth, and seventh generation were 22.35 ± 0.53 h, 17.93 ± 0.72 h, and 18.63 ± 1.1 h, respectively, and the difference between the two groups was not significant (Appendix A). Flow cytometry analysis of expressed surface antigens showed that these cells were uniformly positive for CD44, CD73, CD105, and CD90, and negative for the hematopoietic lineage markers CD45 (Appendix A). No structural or numerical chromosomal abnormalities were found in karyotype analyses of the cells until P14.

### 2.2. HAMC Promotes 3D In Vitro Survival of WJ-MSCs

To determine if HAMC is a suitable delivery vehicle for the injection of WJ-MSCs into the IVD space, we quantified cell viability following 3D culture in 0.5% weight per volume (*w/v*) HAMC and compared it to TissueFill, a commercially available crosslinked hyaluronan hydrogel (Figure 1B). After 7 days, we observed that cell viability was significantly increased with culture in HAMC compared to TissueFill, 52% and 24% viability, respectively (Figure 1B,C).

### 2.3. HAMC Supports Improved Post-Injection WJ-MSCs Viability Compared to TissueFill

The injection of cells for transplantation through a 26-gauge (G) needle can cause cell death due to the shear forces cells will experience as they pass from the syringe to the needle [32,33,34,35], which can be partially negated with the use of a biomaterial for the cell carrier [36]. To determine if WJ-MSC viability is supported by HAMC during the injection process, we recapitulated the cell delivery protocol in vitro and measured acute cell viability following injection (Figure 2A,B). Cells were encapsulated in 0.5% *w*/*v* HAMC, 1% TissueFill, or cell medium and injected through a 26G needle as per the in vivo delivery protocol into a 96-well plastic culture plate (Figure 2A,B). Cell viability was assessed immediately following injection. It was observed that viability decreases significantly following injection in all groups (HAMC, TissueFill, or medium) compared to no injection group, but that cells encapsulated in TissueFill had significantly reduced viability compared to HAMC or cell medium alone (Figure 2C (i) and (ii)). This suggests that HAMC is a more suitable cell delivery vehicle to maintain viability in comparison to TissueFill.

### 2.4. HAMC and WJ-MSCs Transplantation Restores the Disc Anatomy and Water Content of IVDs Following IVD Degeneration

The in-vivo IVD degeneration study was conducted using a well-established rat tail coccygeal needle puncture (Co 6–7 and Co 7–8) model. This model is commonly used due to easy accessibility of this area in rats and reported reproducibility [37]. After 2 weeks after needle injury, we injected PBS, WJ-MSCs, or WJ-MSCs-loaded HAMC (2 × 10^4^ WJ-MSCs in 1 µL + 1 µL of 1% *w/v* HAMC) into the degenerated disc (Co 6–7 and Co 7–8) and performed MRI at six weeks after implantation (Figure 3A). An MRI index was calculated for control disc (no injury), vehicle (PBS treated), HAMC, WJ-MSCs, and HAMC + WJ-MSCs-treated discs (Figure 3B). On coronal T2-weighted MRI, distinct and bright signal intensity of the disc was observed in the control healthy disc (red arrows, Co 5–6) (MRI index: 142.15%). Vehicle-injected disc showed significant decrease in signal intensity and MRI index. However, HAMC/WJ-MSCs-treated discs revealed a significant increase in both signal intensity and MRI index (104.62%) (Figure 3A,B).Whereas, the MRI indices for discs treated with HAMC scaffold or WJ_MSCs only were 39.38% and 61.28%, respectively; almost 3-fold and 2-fold lower, respectively, than the combined injection of HAMC and WJ-MSCs. By contrast, the MRI index for vehicle-treated injured discs was found to be 14.31%, 4-fold lower than that of WJ-MSCs-loaded HAMC-treated discs. Overall, the result indicates that a significantly higher hydration level was retained in HAMC/WJ-MSCs-treated discs compared to HAMC only and WJ-MSCs only implanted discs. Moreover, MRI findings demonstrated that gross anatomical structure and MRI intensity of HAMC + WJ-MSCs injected disc was similar to normal (control) (Figure 3-A), indicating enhanced disc regeneration by HAMC + WJ-MSCs.

### 2.5. HAMC and WJ-MSCs Maintains the Proteoglycan Content and Disc Structure

After MRI analysis, all isolated discs were evaluated by histological analysis. Any abnormalities such as osteophytes formation were absent in all experimental animals. Histological analysis by safranin-O staining and histologic scores demonstrated the preservation of a proteoglycan matrix in the nucleus pulposus (NP) of the disc and histologic structures of the disc. Our results show that Safranin-O-staining intensity of the NP was significantly reduced, and histologic grade was significantly increased after disc injury (Figure 3C (i) and D), indicating decreased proteoglycan content and less preservation of disc structure. By contrast, the HAMC/WJ-MSCs-treated discs showed significantly increased intensity of safranin-O staining and lowest histologic score compared to vehicle, HAMC and WJ-MSCs only injected discs, suggesting that it was able to maintain integrity of disc matrix structure (Figure 3C (i), D and E). In addition to that, HAMC/WJ-MSCs-treated discs revealed a significant increase in aggrecan (a major component of the disc matrix) content in the NP tissue following disc degeneration (Figure 3C (ii) and F). In contrast, dramatically diminished aggrecan expression was observed in vehicle injected discs (Figure 3C (ii) and F) (### *p* < 0.001 (vehicle vs. Sham), ^$^
*p* < 0.05 (HAMC vs. vehicle), * *p* < 0.05 (HAMC/WJ-MSCs vs. WJ-MSCs)). Therefore, the marked histo-morphological difference between HAMC/WJ-MSCs and other groups tested demonstrates the potential of HAMC/WJ-MSCs for viable disc repair.

### 2.6. HAMC and WJ-MSCs Restore the Matrix Proteins and Downregulate Catabolic Enzymes

One of the main consequences of IVD degeneration is the degradation of ECMs such as aggrecan and type II collagen. Herein, extensively less expression of collagen type II (a component of the disc NP matrix) was observed in vehicle-injected discs compared to control discs, whereas, its expression was significantly increased in the HAMC/WJ-MSC-injected discs compared to vehicle, HAMC and WJ-MSC only injected discs (Figure 4A (i), (ii) and B). Immunopositivity for both aggrecan and type II was also higher in WJ-MSCs than vehicle and HAMC, showing efficacy of the cells alone, although it did not reach statistical significance (Figure 4A (i), (ii) and B), ^###^
*p* < 0.001 (vehicle vs. control), ^ *p* < 0.05 (WJ-MSCs vs. HAMC), ** *p* < 0.01 (HAMC + WJ-MSCs vs. WJ-MSCs).

MMP-13 plays a role in the degradation of ECM proteins including aggrecan [11,38]. Representative immunostaining for MMP-13 shows that the percentage of immunopositive cells for MMP-13 was significantly increased in vehicle-treated discs (Figure 5A (i), (ii) and B). In contrast, MMP-13 expression was greatly diminished in HAMC/WJ-MSCs-implanted discs (Figure 5A (i), (ii) and B), ^###^
*p* < 0.001 (vehicle vs. control), ** *p* < 0.01 (HAMC + WJ-MSCs vs. WJ-MSCs).

### 2.7. HAMC and WJ-MSCs Inhibit the mRNA Expression of Pro-Inflammatory Cytokines and Matrix-Degrading Proteases

Next, we evaluated the various cytokines expression in the degenerated discs. As shown in Figure 6, the RT-qPCR results demonstrate significantly upregulated mRNA expression of inflammatory cytokines, inducible nitric oxide synthase (*iNOS*) (Figure 6A) (^###^
*p* < 0.001 vs. control), and matrix-degrading enzymes at 8 weeks following disc injury including: *MMP-13* (Figure 6B) (^###^
*p* < 0.001 vs. control), A disintegrin and metalloproteinase with thrombospondin motifs 4 (*Adamts4*) (Figure 6C) (^#^
*p* < 0.05 vs. control), and Cycloxygenase-2 (*Cox-2*) (Figure 6D) (^#^
*p* < 0.05 vs. control) relative to control. By contrast, mRNA expression of *iNOS* (^***^
*p* < 0.001 vs. vehicle), *MMP-13* (^***^
*p* < 0.001 vs. vehicle), *Adamts4* (^***^
*p* < 0.001 vs. vehicle) and *Cox-2* (^***^
*p* < 0.001 vs. vehicle) was significantly reduced in discs treated with HAMC/WJ-MSCs as compared to vehicle injected discs, indicating that WJ-MSCs-loaded HAMC prevented ECM degradation by suppressing expression of pro-inflammatory cytokines and matrix-degrading enzymes.

## 3. Discussion

Cell therapy for IVD repair has gained considerable interest in the recent decade and direct implantation of MSCs has shown promising therapeutic potential [13]. There are, however, significant challenges, including the fact that the degenerated disc produces a harsh environment consisting of low glucose, low oxygen, low pH due to high lactic acid buildup, and low nutrients combined with an inflammatory milieu [3]. To overcome obstacles for successful disc repair, the selection of appropriate stem cells and the development of biomaterial-based cellular delivery systems that combine cells with cell-carrying matrix molecules such as fibrin and hyaluronan have become increasingly implemented [3]. Recently, various types of biomaterials have been reported to have the ability to prevent cell leakage and promote cell survival and stem cell differentiation toward the IVD cell phenotype [39]. Reports demonstrated that utilization of a carrier such as hyaluronan or annular sealant significantly prevented cell leakage, and the rate of osteophyte formation resulting from cell leakage has been reported to be less than 2.7% in rabbit IVD model [40]. Thus, the use of biomaterials has been strongly recommended to prevent leakage of implanted cells and to promote cell survival and prevent induction of unwanted differentiation of MSCs into osteoblasts [40]. However, the optimal biomaterial for IVD transplantation remains unknown. In our previous study, we demonstrated the efficacy of TissueFill (cross-linked 2% (*w/v*) HA-based dermal filler) as a carrier of WJ-MSCs in a rabbit model of IVD degeneration [4]. TissueFill is commercialized dermal filler and a highly viscous hydrogel solution, but we chose TissueFill as a cell carrier due to the absence of clinically approved biomaterials for the treatment of IVD repair. When cultured in TissueFill, cell viability decreases and injection through a fine-gauge (26G) needle was difficult due to high viscosity and semi-diluted TissueFill was used because less viscous hydrogel was required to increase cell survival and injectability of the biomaterial.

In the recent decade, HAMC has been reported as a promising vehicle for delivery of various types of stem cells in many neurological disease models [24,25,36,41]. However, the HAMC hydrogel has not been tested in animal models of IVD degeneration. In the present study, we investigated the influence of HAMC on cell survival in vitro and thein vivo effects of HAMC loaded with WJ-MSCs in a rat model of IVD degeneration to test potential clinical usability of HAMC for cell delivery. Here, we used HAMC as a cell carrier because the HAMC gel could help prevent cell leakage, enhance cell attachment, exert an anti-inflammatory effect, and provide a favorable micro environment for IVD regeneration [40]. The primary findings of this study are as follows: (1) Our quality control assessment for WJ-MSCs demonstrated that WJ-MSCs showed a spindle fibroblast-like shape until fourteenth generations, which was consistent with our previous report [4]. Flow cytometric analysis of expressed surface antigens showed that these cells were uniformly positive for CD44, CD73, CD105, and CD90, and negative for the hematopoietic lineage markers CD45. Moreover, no structural or numerical chromosomal abnormalities were found in karyotype analyses of the cells until P14; (2) Cell viability in 3D HAMC was significantly greater than in TissueFill, indicating HAMC significantly promotes in-vitro cell survival; (3) Acute cell viability post-injection was significantly higher when cells were injected within an HAMC hydrogel compared to cells injected in TissueFill, suggesting that HAMC is more suitable for delivery of WJ-MSCs; (4) A single intradiscal injection of WJ-MSCs-loaded HAMC resulted in enhanced IVD regeneration compared with implantation of WJ-MSCs or HAMC alone, confirmed by radiological and histological findings—WJ-MSC-loaded HAMC showed the best regenerative effect by inhibiting pro-inflammatory cytokines and matrix degenerative enzymes.

In the in vitro cell viability assay, we observed a drop in cell viability following injection through a syringe with both HAMC and TissueFill; however, cell viability was improved compared to injection in PBS. This acute decrease was likely caused by the shear stress experienced by cells as they passed through the fine-gauge needle, as has been reported elsewhere [32,33,34,35], although the process of dissociating WJ-MSCs during the injection preparation may have also contributed. We also observed a decrease in WJ-MSC viability after 7 days of culture in HAMC or TissueFill, when compared with 2D standard culture, although the viability in HAMC was significantly greater than in TissueFill. The cells were cultured in the hydrogels in suspension with no additional media or supplements in order to challenge the cells with stress, which may have contributed to their decrease in viability over time. Furthermore, HAMC is reported to be approximately 90% degraded in vitro at 14 days [42], and therefore it is likely that by day 7 the mechanical strength of HAMC and TissueFill is reduced which may lead to reduced cell support.

In this study, we injected human WJ-MSCs into the rat xenograft model, but did not administer immunosuppressive agents based on the fact that WJ-MSCs has the weakest expression of immune related genes, including MHC II genes, Toll-like receptor 4 (TLR4), TLR3, Jagged1 (JAG1), neurogenic locus notch homolog protein 2 (NOTCH2), and NOTCH3 compared to MSCs isolated from other sources [40]. Our study showed that the HAMC/WJ-MSCs-implanted discs showed an overall decrease in inflammation of the disc. The HAMC hydrogel has been shown to reduce the inflammatory response in the CNS [26]. For example, intrathecal injection of the HAMC decreased the levels of IL-1α in injured spinal cord due to anti-inflammatory properties of HA [26]. It has been also reported that the HA itself acts as an anti-inflammatory molecule by downregulating the expression of proinflammatory cytokines such as IFN-γ and the apoptosis marker caspase-3 [43]. Our data demonstrate that the HAMC alone injected disc showed attenuated mRNA expression of *iNOS, MMP-13, Adamts4*, and *Cox-2* and increased expression of aggrecan compared to vehicle alone injected disc (Figure 5 and Figure 6), suggesting that combined injection of WJ-MSCs and HAMC promoted regenerative potential in degenerated disc.

Two key points to be addressed in future work include (1) by what mechanism implanted WJ-MSCs modulate the inflammatory microenvironment and vice versa, and (2) whether implanted WJ-MSCs would differentiate into NP cells. By incorporation of WJ-MSCs into a new biomaterial, HAMC could enhance cell survival by modulating inflammatory microenvironment and promote the ability to differentiate into the NP cell-like phenotype. In the present study, however, we did not track the implanted WJ-MSCS and could not confirm survival of implanted WJ-MSCS at 6 weeks post implantation. In terms of survival of implanted cells, there have been several reports on long-term survival (more than 3 months) of implanted MSCs loaded into biomaterials [44,45]. In our previous study, WJ-MSCs-loaded TissueFill (1% cross-linked hyaluronic acid hydrogel) was intradiscally implanted into degenerated rabbit disc and no survived cells were found 12 weeks post implantation [4]. Here, we confirmed that a complex mixture of cytokines produced by WJ-MSCs, including TGF-βligands (TGFβ1, TGFβ2, and TGFβ3), growth differentiation factor-15 (GDF-15), chemokine (C-C motif) ligand 5 (CCL5), and MMP1, could trigger the multiple signaling system including TGF-β signaling and stimulate IVD regeneration. WJ-MSCs are capable of immune modulation, immune suppression, and immune avoidance, making them optimal candidates for cellular therapies in allogenic transplantation [20]. WJ-MSCs have been extensively investigated as an anti-inflammatory agent, which is associated with their remarkable immunomodulatory effects [46,47]. Moreover, MSCs have been proven to be associated with suppression of lymphocyte activation. Similarly, MSCs express inducible nitric oxide synthase (iNOS) and indoleamine 2,3-dioxygenase (IDO) which leads to down-regulation of T cell proliferation in rodent and human MSCs, respectively [24,47,48]. To date, the anti-apoptotic property of stem cells seems to be the most widely established beneficial effect of MSCs [47,49]. In an earlier study, administration of MSCs displayed a renoprotective effect by preventing tissue apoptosis in an acute kidney injury (AKI) model. Briefly, MSC-treated AKI mice demonstrated the increased expression of the anti-apoptotic gene BCL2 and significant downregulation of the pro-apoptotic gene BAX [50].

Hence, the regenerative effects of HAMC/WJ-MSCs in our study might therefore be linked with immunomodulatory and anti-inflammatory effects via paracrine signaling, regardless of whether WJ-MSCs differentiated into NP-like cells. In addition, HAMC provided a superior carrier for WJ-MSCs and improved regeneration in a rat model of disc degeneration.

To the best of our knowledge, no other study has investigated the combined treatment of WJ-MSCs and HAMC hydrogel, to date. Previously, we investigated the efficacy of combined implantation of WJ-MSCs and TissueFill in a rabbit model of disc degeneration [4]. Similarly, we conducted a Phase 1 clinical trial by applying combined transplantation of autologous adipose-derived MSCs and TissueFill in chronic discogenic LBP patients: six patients among 10 participants showed improvement of pain and disability [1]. Herein, we have avoided the highly viscous nature of TissueFill by employing HAMC as an alternative, and based on our results, HAMC hydrogel was found to be superior to TissueFill, however, the explicit mechanism by which disc structure was restored remains to be investigated. It would be valuable to compare the in-vivo effects between TissueFill and HAMC hydrogel in future studies. Additionally, the regenerative effects of WJ-MSCs loaded HAMC must be tested using larger animal models to understand the feasibility of applying this approach to humans. Despite some limitations, this investigation demonstrates that regeneration following disc degeneration can be achieved by combined implantation of WJ-MSCs and HAMC.

## 4. Materials and Methods

### 4.1. Preparation of Hydrogels

Hyaluronan (HA, 1200–1900 kDa, Novamatrix, Drammen, Norway) and methylcellulose (MC, 300 kDa, Shin-Etsu, Tokyo, Japan) (Figure 1A) was used to prepare an injectable, biocompatible, and biodegradable hydrogel scaffold HAMC, as previously described [41]. Briefly, 24 hours before use, sterile-filtered HA and MC were dissolved into artificial cerebrospinal fluid (aCSF; R&D Systems, Minneapolis, MN, USA), cell medium, or PBS at a concentration of 1% *w/v* and left at 4 °C overnight. Prior to use, HAMC was kept on ice for all experiments. TissueFill, a cross-linked HA commercially available hydrogel, was purchased and diluted to a 1% solution in aCSF or cell medium for use in in vitro experiments.

### 4.2. Preparation and Quality Control of WJ-MSCs

WJ-MSCs were purchased from the Good Manufacturing Practice (GMP) facility of CHA Biotech (Seongnamsi, Korea) and this study was conducted according to the World Medical Association Declaration of Helsinki. With the written informed consent and ethics committee approval (No. BD2014-055), umbilical cord was obtained immediately after birth.

WJ-MSCs were prepared as described elsewhere [4] and all culture conditions adhere to GMP standards. Briefly, the cord was washed with phosphate-buffered saline (PBS) and Wharton’s jelly was then cut into pieces smaller than 5 mm^3^ after blood vessels were removed. The minced Wharton’s jelly was digested for 6–10 h in a sterilized bottle with 15 mL culture medium containing collagenase of type I at 0.075% in 5% carbon dioxide, at 37 °C with agitation in an incubator. The cells were then washed three times with D-Hank’s salt solution and centrifuged at 250× *g* for 10 min at room temperature, and the cells were resuspended in culture medium (Dulbecco’s modified Eagle’s medium; DMEM) with low glucose (DMEM-L; Gibco BRL) supplemented with 10% (*v/v*) fetal bovine serum (FBS; Gibco BRL) and 1% antibiotic–antimycotic solution (Gibco BRL) in humidified air with 5% carbon dioxide at 37 °C. The cells were harvested after reaching 70–80% confluency and characterization tests were performed at passage 7. Quality control of these cells was performed according to the standards of the Korea’s Ministry of Food and Drug Safety.

### 4.3. In Vitro 3D Hydrogel Cell Survival

WJ-MSCs were cultured in a serum-free Dulbecco’s modified Eagle’s medium (DMEM; Gibco BRL, Gaithersburg, MD, USA). On the day of the experiment, WJ-MSCs were dissociated using trypsin and 1 × 10^4^ cells were suspended into either 150 uL of 0.5% *w/v* HAMC in medium or 1% TissueFill or seeded in 2D with cell medium and pipetted into a 96-well plastic culture plate. Cells were cultured for 0, 1, 3, or 7 days and viability was assessed using calcein AM (for live cells) and ethidium homodimer (for dead cells) (ThermoFisher Waltham, MA, USA). Stained cells were imaged on a Zeiss 880 fluorescence microscope (Germany), and three fields of view per well were captured as a z stack approximately 1mm in depth for the 3D hydrogel groups. The z stack was merged into a single image and the number of live cells per field of view was determined using ImageJ (La Jolla, CA, USA) and averaged across three wells within each biological replicate, with three replicates in total. Cell viability was expressed as the percentage of live cells at each time point normalized to the initial cell viability at time 0.

### 4.4. In Vitro Post-Injection Cell Viability

Wharton’s jelly-derived mesenchymal stem cells (WJ-MSCs) were cultured in a serum-free Dulbecco’s modified Eagle’s medium (DMEM; Gibco BRL, Gaithersburg, MD, USA). On the day of the experiment, WJ-MSCs were trypsinized and 2 × 10^4^ cells/uL were suspended into equal volume of 1% HAMC, final concentration 0.5% HA/ 0.5% MC, *w/v*, 1% TissueFill, or cell medium and loaded into a Hamilton 10 µL Gastight syringe #1701 fitted with a 26 G 45° beveled needle. Injections were performed to mimic the in vivo cell delivery protocol by injecting 50 uL into a 96-well plate in triplicate. Immediately following injection, 200 uL of cell medium was pipetted on top of the cells. To assess post-injection cell viability, 20 uL of CCK-8 (Sigma-Aldrich, St Louis, MO, USA) was added to each well and incubated for 90 minutes. A spectrophotometer was used to measure optical density at 450 nm of each well to determine relative cell viability. Viability was expressed as optical density normalized to viability at time 0.

### 4.5. Injury-Induced Disc Degeneration Rat Model

The animal experiments were performed as directed and approved by the Institutional Animal Care and Use Committee (IACUC) of CHA Bundang Medical Center (IACUC180189)**.** Six-week-old female Sprague-Dawley rats (220–240 g) were purchased from Orient Bio Inc., Korea, and were acclimatized for a week at a life/dark cycle of 12/12 h (temperature; 22 ± 1 °C and relative humidity; 50% ± 1%) and free access to food and water ad libitum.

Prior surgery, rats were deeply anesthetized with general anesthesia mixture of Zoletil^®^ (50 mg/kg Virbac Laboratories, France) and Rompun^®^ (10 mg/kg, Bayer, Korea) injected intraperitoneally. Thereafter, the proximal most part of the tail along with the pelvic area was sterilized with 70% alcohol followed by povidone iodine. A 1-cm longitudinal incision was made along the tail exposing the lateral portion of the coccygeal disc. Following that, a 21G sterile needle was inserted into the center of the coccygeal discs (Co6-7, Co7-8). To prevent further damage, needle insertion was limited to 5 mm in depth. Following which, spinal needle was rotated at complete 360 degree and kept in place for 30 s. Finally, the skin was sutured, disinfected and as a prophylactic intervention, 0.9% sterile normal saline (5 mL) was injected subcutaneously, and appropriate dose of analgesic (Ketoprofen, SCD Pharm. Co. Ltd., Korea) and antibiotic (Cefazolin, CKD Pharmaceuticals, Korea) for 3 days after surgery. During procedure, body temperature of rats was maintained at 37 °C with the help of thermostatically regulated heating pad.

### 4.6. Experimental Design in a Rat Disc Degeneration Model

Our data also showed that WJ-MSCs maintained high viability and pluripotency without any karyotype abnormalities until the nineteenth passage (Appendix A). The seventh passage of WJ MSCs was therefore used for our study.

Forty rats were randomly divided into four groups: (1) PBS vehicle, (2) HAMC, (3) WJ-MSCs (2 × 10^4^ cells), and (4) WJ-MSCs/ HAMC (*n* = 10/each group) treated groups. Thenormal discs (Co5-6) proximal to the treated discs in each tail were used as a control. Two weeks following disc injury, ten animals per group received either PBS only (vehicle group), WJ-MSCs (seventh passages, 2 × 10^4^ cells/2 µL/disc), referred to as WJ-MSC group, HAMC (2 µL 1% HAMC with PBS = 0.5% HAMC) and WJ-MSCs-loaded HAMC (1 µL of 2 × 10^4^ cells + 1 µL 1% HAMC = 0.5% HAMC (HAMC/WJ-MSCs group). Coccygeal discs were removed following sacrifice 6 weeks after implantation for radiologic and histologic analysis.

### 4.7. Magnetic Resonance Imaging (MRI)

Six weeks after implantation, we performed 9.4 T MRI (Bruker BioSpec, USA) to compare degree of coccygeal disc degeneration and water content of the disc. T2-weighted imaging protocol was set as: (1) coronal plane; time to repetition (TR) of 5000 ms, time to echo (TE) of 30 ms, 150 horizontal × 150 vertical matrix; field of view of 15 horizontal × 15 vertical, and 0.5 mm slices with 0 mm spacing between each slice. (2) Sagittal plane; time to repetition (TR) of 5000 milliseconds (ms), time to echo (TE) of 50 ms, 200 horizontal × 600 vertical matrix; field of view of 20 horizontal × 60 vertical, and 0.8 mm slices with 0 mm spacing between each slice. The signal intensity and MRI index (the area of NP multiplied by average signal intensity) were evaluated [51]. The region of interest (ROI) was defined as high signal intensity area in the mid coronal plane of the T2-weighted images; as the outline of the NP, measured the ROI using Image J software (the National Institutes of Health, Bethesda, MD, USA) [51]. The measurement of MRI index was performed by two independent observers who were blinded to the specimen’s treatment.

### 4.8. RNA Isolation and Real Time RT-PCR

Six weeks after implantation, the coccygeal discs were harvested and approximately 100 mg NP was isolated from the disc and triturated under liquid nitrogen in a pre-cooled mortar. Liquid nitrogen was allowed to volatilize and with the help of pestle, hardened nucleus pulposus was ground into a powder. Viscous NP was then incubated in 1 mL Trizol (Invitrogen, Carlsbad, CA, USA) at 37 °C for 10 min. Chloroform (0.2 mL) was then added, and centrifuged at 12,000 *g* and 4 °C for 15 min. Following centrifugation, the top layer was transferred to 0.5 mL isopropanol for precipitation and centrifuged again. Next the supernatant was removed, and RNA precipitate was washed once with 75% alcohol and dried at 37 °C for 10 min. Then the RNA was dissolved with 20 µL RNase-free water. For PCR analysis, 1 µg of RNA was reverse transcribed to complementary DNA (cDNA) using an RT PreMix Kit (Seoul, Korea). Real time PCR was performed by using SYBR green master mix and mRNA expression was analyzed by ABI Step one Real-time PCR system (Applied Biosystems, UK). Primer sequences for the gene of interest used in this study are given in Table 1. The typical PCR amplification profile used was denaturation at 95 °C for 10 min followed by a second step at 95 °C for 15 s, followed by annealing and extension at 60 °C for 30 s and melting curve analysis at 40 cycles, in which dissociation curve software was used to ensure that only a single product was amplified. Target genes were normalized with glyceraldehyde 3-phosphate dehydrogenase (GAPDH) and data were analyzed by 2^−ΔΔct^ method.

### 4.9. Safranin-O Staining

Six weeks after implantation, rats were euthanized and discs from each rat were harvested for histological analysis. Following which, discs with adjacent vertebral body were fixed in 10% neutral buffered formalin for one week, decalcified in RapidCal Immuno (BBC Biochemical, Mount Vernon, WA, USA) for 2 weeks. Tissues were then processed for paraffin embedding and sectioning into coronal sections (10 µm) using microtome (Leica). The sections were dewaxed, rehydrated, and stained with Safranin-O (Sigma, USA) to evaluate the quantity and distribution range of proteoglycans. Then, sections were mounted using mounting media and scanned with an OLYMPUS C-mount camera adapter (U-TVO.63XC, Tokyo, Japan). All samples were subjectively assessed for tissue morphology and architecture by pathologists who were blinded to the sample information. For the assessment of histologic structure, a 14-point score was used based on safranin-O staining [52]. At 6 weeks post-implantation, the scoring consists of five parameters: NP structure, NP clefts/fissures, annulus fibrosus (AF)/NP boundary, AF structure, and AF/clefts/fissures. The sum of the separate scores ranges from 0 (normal) to 14 (most severe) [52].

### 4.10. Immunohistochemistry

Six weeks following implantation, rats were euthanized via carbon dioxide inhalation and coccygeal discs were collected, and immunohistochemical analysis was performed for aggrecan, collagen type II, and MMP-13. Briefly, explants were fixed overnight in a 4% paraformaldehyde (PFA) solution and decalcified in a decalcification solution; RapidCal Immuno (BBC Biochemical, Mount Vernon, WA, USA) for 2 weeks. Thereafter, discs were embedded within paraffin wax and sectioned longitudinally using a microtome (Leica) into 5- to 10-μm thickness sections. Prior to immunohistochemical staining, sections were dewaxed, rehydrated, and stained with antibodies against aggrecan (1:1000, Abcam, UK), collagen type II (1:100, Abcam, UK) and MMP-13 (1:200, Abcam, UK). After 24 hours, sections were washed with phosphate buffered saline with Tween 20 and incubated with the secondary antibody anti-Rb horseradish peorixdase (Roche Diagnostics Ltd., Switzerland), and Alexa 488-conjugated secondary antibodies (Invitrogen, USA). After a washing step, the specimens were counterstained with DAPI (1:500, Abcam, UK) for 10 min. Finally, the sections were mounted and examined using a fluorescence microscope (Zeiss 880, Germany and Leica SP5, Germany). The number of positive cells was counted in three random fields (*n* = 4 per group) (×40) using Image J software (https://imagej.nih.gov/ij/).

### 4.11. Statistical Analysis

Data were analyzed by using GraphPad Prism (version 5.01, GraphPad Software). Data are presented as mean ± standard error of the mean (SEM) unless otherwise stated. In vitro cell viability data were analyzed by two-way ANOVA, followed by Tukey’s post-hoc test *p* < 0.05. PCR and immunofluorescence data were analyzed using one-way ANOVA, followed by Tukey’s post-hoc test. *p*-values < 0.05 were considered statistically significant.

## 5. Conclusions

In conclusion, this study addresses the efficacy of combined implantation of WJ-MSCs and HAMC for IVD regeneration. Combined injection of WJ-MSCs and HAMC enhanced IVD regeneration through increase in cell survival, attenuation of the activation of iNOS, MMP-13, ADAMTS4 and COX-2, and significant up-regulation of ECM, such as aggrecan and collagen type II. This strategy offers an advantage for clinical application because WJ-MSCs-loaded HAMC induces IVD repair more effectively than cell injection alone and supports the potential clinical use of HAMC for cell delivery to attenuate IVD degeneration and promote regeneration.

## Figures and Tables

**Figure 1 ijms-21-07391-f001:**
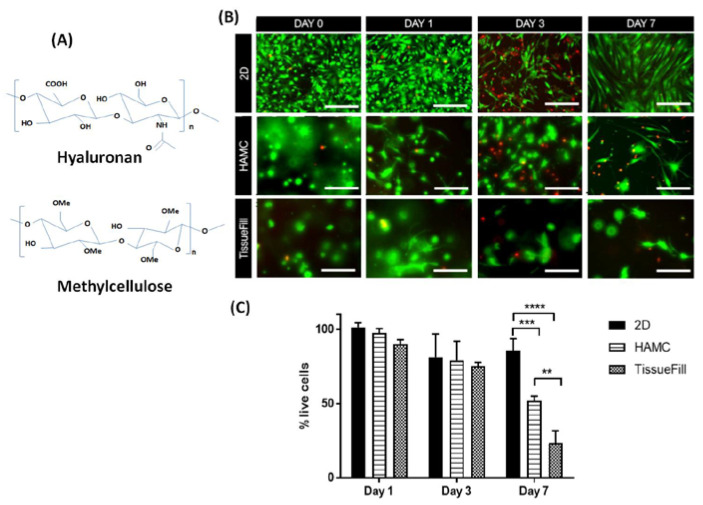
HAMC supports 3D WJ-MSCs survival in vitro. Cells were cultured in 3D HAMC and compared to 2D plastic culture or 3D TissueFill. Cell viability was significantly higher for cells cultured in HAMC compared to TissueFill. (**A**) Molecular structure of hyaluronan (HA) and methyl cellulose (MC). (**B**) Representative immunofluorescence images for live (green) and dead (red) assay in 2D, HAMC and TissueFill at day 0, day 1, day 3 and day 7 following WJ-MSCs culture. (**C**) Quantitative bar graph illustrating the percentage of live cells in 2D, HAMC and TissueFill at day 0, day 1, day 3 and day 7 following WJ-MSC culture. After 7 days of culture HAMC had significantly higher viability than TissueFill. Mean + SD, *n* = 3 biological replicates. Two-way ANOVA with Tukey’s post hoc, ** *p < 0.01, *** p < 0.001, **** p < 0.0001*. Scale bar = 50 µm.

**Figure 2 ijms-21-07391-f002:**
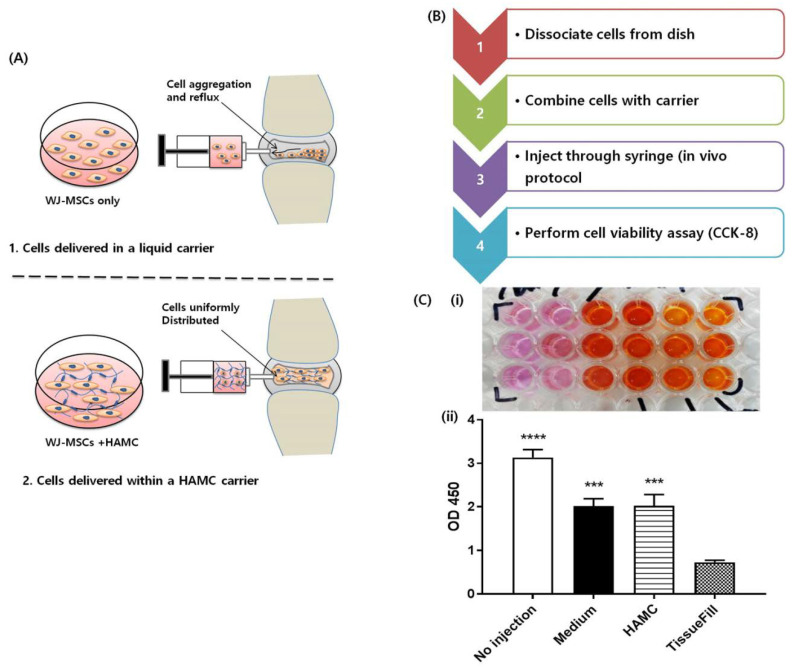
Encapsulation in HAMC maintains post-injection viability compared to TissueFill. WJ-MSCs were encapsulated in different vehicles and injected through a fine-gauge needle to assess post-injection viability. (**A**) Schematic illustrating the comparison of WJ-MSC stability and viability between HAMC-encapsulated, and non-encapsulated WJ-MSCs post-injection. (**B**) Stepwise post-injection procedure. (**C**) (i) Representative CCK-8 assay, (ii) quantitative result after CCK-8 assay, compareing the cell viability of WJ-MSCs in medium alone, TissueFill or HAMC. Cells injected in medium or HAMC retained significantly higher viability compared to TissueFill. Mean + SD, *n* = 3 biological replicates. One-way ANOVA with Tukey’s post hoc, *** *p* < 0.001, **** *p* < 0.0001 compared to TissueFill group.

**Figure 3 ijms-21-07391-f003:**
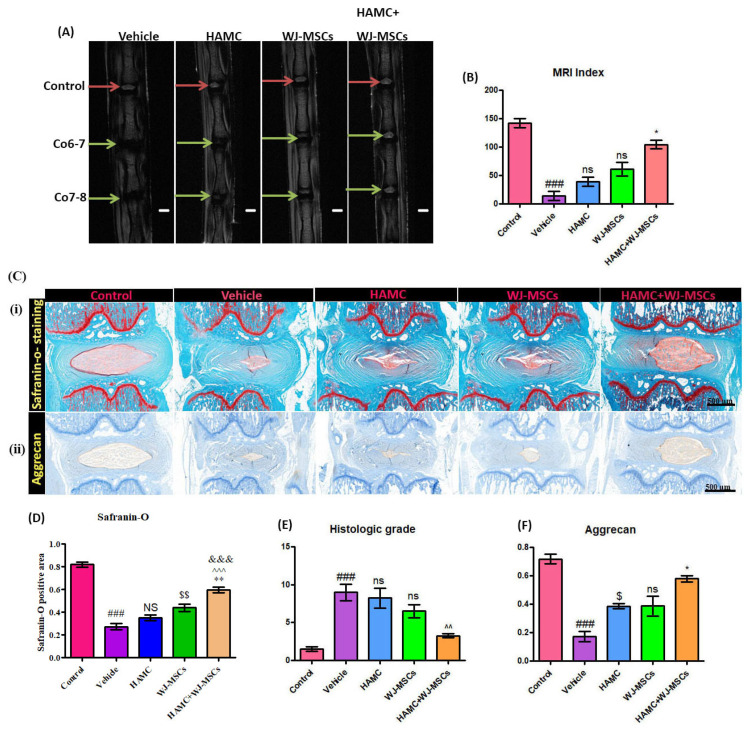
WJ-MSCs-loaded HAMC injection promotes repair of the degenerated disc (**A**) Representative T2-weighted MRI of the coccygeal discs: control, vehicle-treated, HAMC-treated, WJ-MSCs only treated, and HAM + CWJ-MSC-treated disc (coronal view) (red arrow = control, green arrow = needle-punctured disc). (**B**) MRI index for discs of each group. Mean ± SEM, (*n* = 3), one-way ANOVA followed by Tukey’s post-test. ## *p* < 0.05 (vehicle vs. control), * *p* < 0.05 (HAMC + WJ-MSCs vs. WJ-MSCs); ns = non-significant. MRI performed at 6 weeks after implantation showed the best restoration of disc anatomy and water content of the disc in HAMC/WJ-MSC-implanted discs. (**C**) (i) and (**D**) Representative images of safranin-O staining and quantitative safranin-O positive area, revealed the highest preservation of disc structure in HAMC/WJ-MSCs-injected disc (×200). Mean ± SEM, (*n* = 3), one-way ANOVA followed by Tukey’s post-test. ^###^
*p* < 0.001(vehicle vs. Sham), ^$$^
*p* < 0.01 (WJ-MSCs vs. HAMC), ** *p* < 0.01 (HAMC + WJ-MSCs vs. WJ-MSCs), ^^^ *p* < 0.001 (HAMC + WJ-MSCs vs. HAMC), and &&& *p* < 0.001 (HAMC + WJ-MSCs vs. vehicle). (**E**) The quantitative histological score and (**C**) (ii) and (**F**) immunoreactivity for aggrecan showed the highest expression in HAMC/WJ-MSCs-implanted discs. Mean ± SEM, (*n* = 3), one-way ANOVA followed by Tukey’s post-test. ^###^
*p* < 0.001(vehicle vs. Sham), ^$^
*p* < 0.05 (HAMC vs. vehicle), * *p* < 0.05 (HAMC + WJ-MSCs vs. WJ-MSCs).

**Figure 4 ijms-21-07391-f004:**
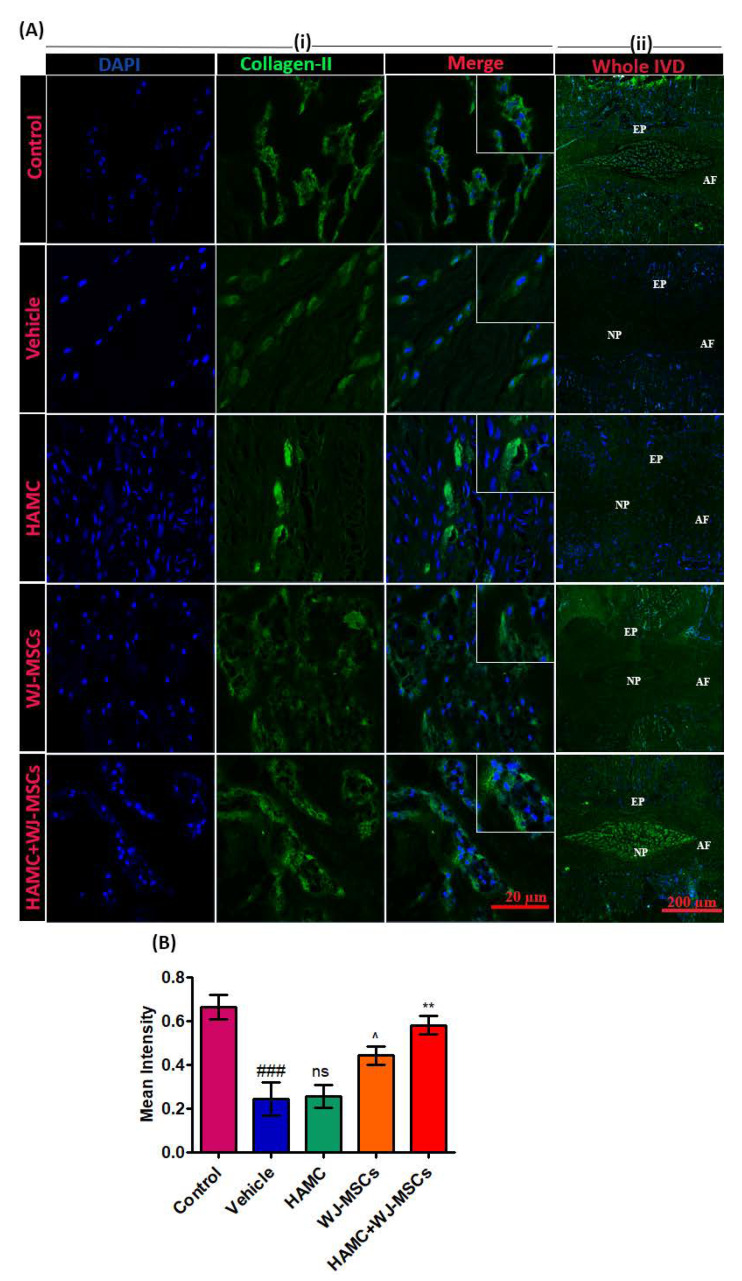
Combined injection of HAMC and WJ-MSC significantly restores collagen type II in degenerated disc. (**A**) Representative collagen II immunofluorescent staining. (i) Collagen expression at region of interest (ROI) and (ii) full power images demarcating entire disc (nucleus pulposus—NP, annulus fibrosus—AF and end plate—EP), and collagen II expression in whole NP area. (**B**) Quantitative immunofluorescence intensity for collagen II expression. Mean ± SEM, (*n* = 3), one-way ANOVA followed by Tukey’s post-test. ^###^
*p* < 0.001 (vehicle vs. control), ^ *p* < 0.05 (WJ-MSCs vs. HAMC), ** *p* < 0.01 (HAMC + WJ-MSCs vs. WJ-MSCs), ns = non-significant. Combined injection of HAMC and WJ-MSCs significantly preserved the extracellular matrix (ECM) content, collagen type II compared to the vehicle, HAMC, or WJ-MSCs only injected discs.

**Figure 5 ijms-21-07391-f005:**
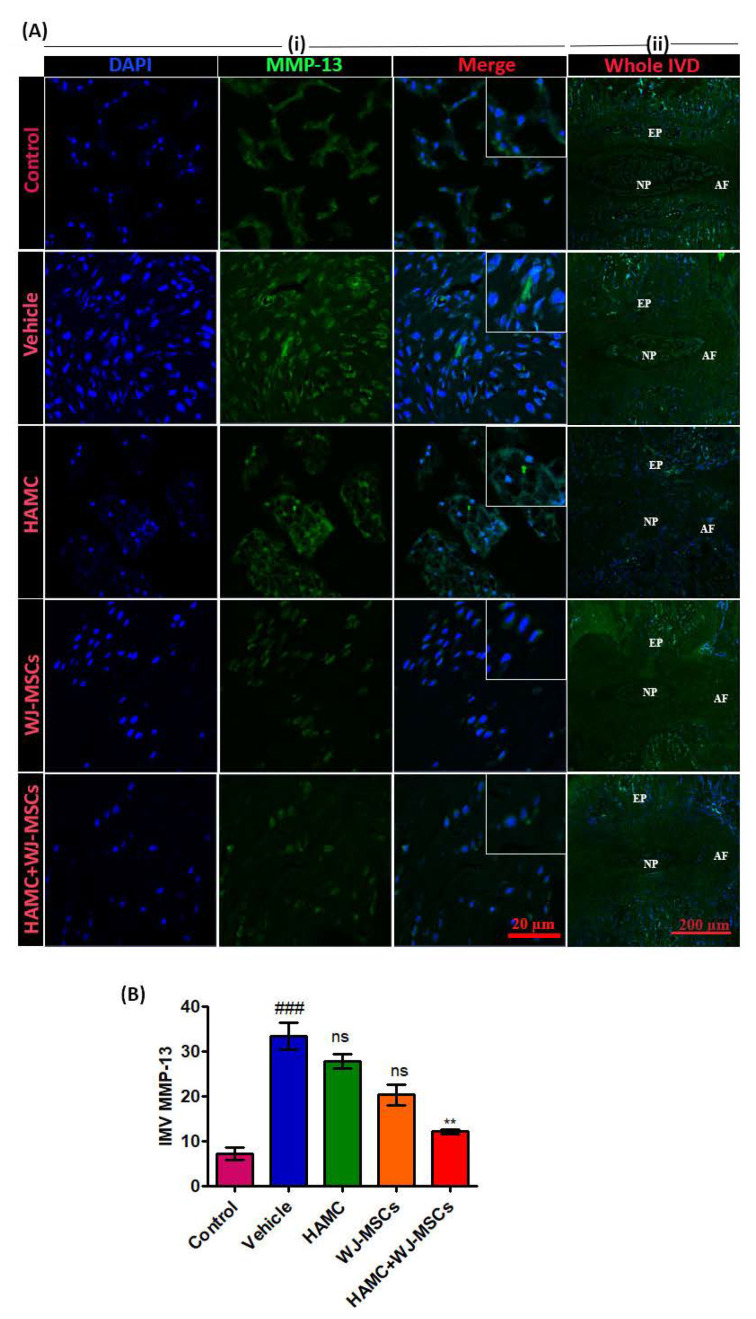
Combined injection of HAMC and WJ-MSC substantially reduced matrix metalloproteinase-13 (MMP-13) expression in degenerated disc. (**A**) Representative MMP-13 protein levels as determined by immunofluorescence staining. (i) MMP-13 expression (green) at region of interest (ROI), DAPI (blue) and merge signal (ii) full power images (far right panel) demarcating entire disc (nucleus pulposus—NP, annulus fibrosus—AF and end plate—EP), and MMP-13 expression in whole NP area. (**B**) Quantitative immunofluorescence intensity for MMP-13 expression. Immunopositivity was counted in at least three random low power fields (×40) and calculated as mean intensity. Mean ± SEM, (*n* = 3), one-way ANOVA followed by Tukey’s post-test. ^###^
*p* < 0.001 (vehicle vs. control), ** *p* < 0.01 (HAMC + WJ-MSCs vs. WJ-MSCs). Ns = non-significant. We found that combined injection of HAMC and WJ-MSC significantly diminished the expression of matrix-degrading enzyme, MMP-13.

**Figure 6 ijms-21-07391-f006:**
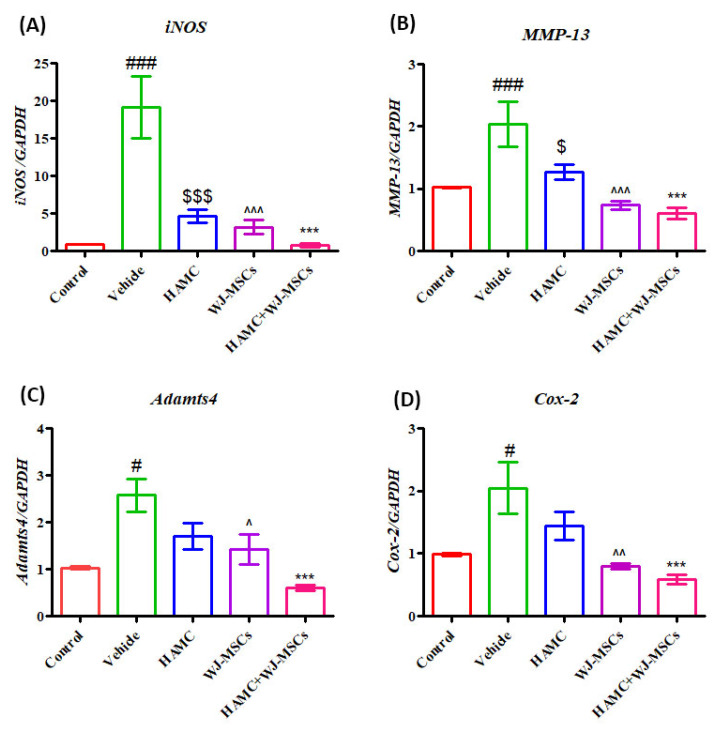
Combined injection of HAMC and WJ-MSC inhibits the mRNA expression of pro-inflammatory cytokine and matrix-degrading enzymes. (**A**) *iNOS* mRNA expression, (**B**) *MMP-13* mRNA expression, (**C**) *Adamts4* mRNA expression, and (**D**) *Cox-2* mRNA expression. In the HAMC/WJ-MSCs-injected discs, mRNA expression of pro-inflammatory cytokine (inducible nitric oxide synthase; *iNOS*) and matrix-degrading enzymes, including *MMP-13*, A disintegrin and metalloproteinase with thrombospondin motifs 4 (*Adamts4*) and Cycloxygenase-2 (*Cox-2*) were found to be significantly downregulated compared to the vehicle-injected disc. Data are presented as mean ± SEM, one-way ANOVA followed by Tukey’s post-hoc test. (*n* = 3). For time, glyceraldehyde 3-phosphate dehydrogenase *(GAPDH*) was used as internal control. ^#^
*p* < 0.05, ^##^
*p* < 0.01, ^###^
*p* < 0.001 (vehicle vs.control), ^$^
*p* < 0.05, ^$$$^
*p* < 0.001 (HAMC vs. vehicle), ^^^
*p* < 0.05, ^^^^^
*p* < 0.001 (WJ-MSCs vs. vehicle only), * *p* < 0.05, ** *p* < 0.01, *** *p* < 0.001 (HAMC + WJ-MSCs vs. vehicle only).

**Table 1 ijms-21-07391-t001:** Primer sequences for references and target genes.

Primers	Direction	Sequences
*Adamts4*	Forward	5′-CGTGGTGTGTGTGTGTGT-3′
	Reverse	5′- AGAGGAAAGTAGGGCAGGT-3′
*Cox-2*	Forward	5′-TGTATGCTACCATCTGGCTTCGG-3′
	Reverse	5′-GTTTGGAACAFTCGCTCGTCATC-3′
*MMP-13*	Forward	5′-TGGTCCCTGCCCCTTCCCTA-3′
	Reverse	5′-CCGCAAGAGTCACAGGATGGTAGTA-3′
*iNOS*	Forward	5′-CTGCAGGTCTTTGACGCTCGGAG -3′
	Reverse	5′-GTGGAACACAGGGGTGATGCTCC-3′
*GAPDH*	Forward	5′-CAACTCCCTCAAGATTGTCAGCAA-3′
	Reverse	5′-GGCATGGACTGTGGTCATGA-3′

Inducible nitric oxide synthase, *iNOS*; matrix metalloproteinase-13, *MMP-13*; A disintegrin and metalloproteinase with thrombospondin motifs 4, *Adamts4*; Cycloxygenase-2, *Cox-2;* glyceraldehyde 3-phosphate dehydrogenase, *GAPDH*.

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
