# Peer review of "An Injectable Hyaluronan–Methylcellulose (HAMC) Hydrogel Combined with Wharton’s Jelly-Derived Mesenchymal Stromal Cells (WJ-MSCs) Promotes Degenerative Disc Repair"

_ijms, 2020, doi:10.3390/ijms21197391_

Round 1
Reviewer 1 Report
Dear Authors,
The submitted manuscript entitled "An injectable hyaluronan-methylcellulose (HAMC) hydrogel combined with Wharton’s jelly-derived mesenchymal stem cells (WJ-MSCs) promotes degenerative disc repair" compromise a very valuable research in the field. However a number of corrections must be performed, to improve further the quality.
1) 1. Introduction, lines 47-97. The introduction should also focus more on the possible applications of WJ-MSCs in regenerative medicine and tissue engineering. Also the unique properties of WJ-MSCs as immonoregulatory cells must be attributed. To this direction you should seek valuable information in the publication Mesenchymal stromal cells as potential immunomodulatory players in severe acute respiratory distress syndrome induced by SARS-CoV-2 infection. World J Stem Cells 12(8):731-751. doi: 10.4252/wjsc.v12.i8.731.
Also, the authors should appropriate define the cells that they intended to use in the current study based on the recent publication of MSCs commitee of ISCT (Viswanathan S, Shi Y, Galipeau J, Krampera M, Leblanc K, Martin I, Nolta J, Phinney DG, Sensebe L. Mesenchymal stem versus stromal cells: International Society for Cell & Gene Therapy (ISCT®) Mesenchymal Stromal Cell committee position statement on nomenclature. Cytotherapy 2019; 21: 1019-
1024 [PMID: 31526643 DOI: 10.1016/j.jcyt.2019.08.002]). Should be Mesenchymal Stem Cells or Stromal cells.
2) 5. Methods - 5.2 In vitro 3D hydrogel cell survival, lines 277-288. The authors should report the origin of WJ-MSCs. These cells were commercially available or provided by another laboratory. Due to Bioethics resrtictions, signed informed consent by the mothers should be reported in this section. Also, the informed consent should be according the declaration of Helsinki.
3) 5. Methods - 5.3 In vitro post-injection cell viability, lines 289-300
Quality control assessment of the WJ-MSCs, should be performed by the authors and the results could be added either in the main manuscript or in supplementary section.
4) 5. Methods - 5.5 Experimental design in rat disc degenration model, lines 319 - 327. The authors mentioned that for their experiments they used WJ-MSCs of passage 7. In this passage some morphological features and characteristics may be changed. The evaluation of the quality characteristics of WJ-MSCs should also be extended in this passage.
5) 5. Methods - 5.7 RNA isolation and real time RT-PCR. The authors evaluated the expression of ADAMTS4, COX-2, MMP-13, and iNOs. However, further evaluation in proteomic level (e.g. Western Blot) should be performed, in order the results to be accurate. Also according to genes nomenclature, if the above genes are reffered to rats should be written Adamts4, Cox-2 and etc.
6) In addition, the authors should discuss the new results and try to compare their results with other related studies.
7) 3. Discussion, lines 247 - 248. Please remove the term first from the following sentence. To the best of our knowledge, this study is the first...HAMC hydrogel. Do the same in line 261.
Thank you in advance.
Author Response
Dear Ms. Nikolina Popovic,
We would like to thank you and the reviewers for considering our manuscript “An injectable hyaluronan-methylcellulose (HAMC) hydrogel combined with Wharton’s jelly-derived mesenchymal stem cells (WJ-MSCs) promotes degenerative disc repair” (ijms-939826) and providing us the valuable feedback. We have revised our manuscript according to the reviewer’s comments as detailed below. Moreover, a list of changes is highlighted in red in the revised manuscript.
Response to Reviewer:
Comments#1:The submitted manuscript entitled "An injectable hyaluronan-methylcellulose (HAMC) hydrogel combined with Wharton’s jelly-derived mesenchymal stem cells (WJ-MSCs) promotes degenerative disc repair" compromise a very valuable research in the field. However a number of corrections must be performed, to improve further the quality.
1) 1. Introduction, lines 47-97. The introduction should also focus more on the possible applications of WJ-MSCs in regenerative medicine and tissue engineering. Also the unique properties of WJ-MSCs as immonoregulatory cells must be attributed. To this direction you should seek valuable information in the publication Mesenchymal stromal cells as potential immunomodulatory players in severe acute respiratory distress syndrome induced by SARS-CoV-2 infection. World J Stem Cells 12(8):731-751. doi: 10.4252/wjsc.v12.i8.731.
Response#1: We greatly appreciate the reviewer’s suggestion. As advised, we included following part in introduction of the revised manuscript (line72-92)
Despite some positive outcome of MSCs-based therapies, the strength of the evidence for use of MSCs in IVD degeneration is low, due to the significant risk of bias, small sample size, and lack of a control group in clinical trials[1]. Furthermore, for the successful development of MSCs-based cell therapy for disc repair, there are still many obstacles to overcome such as the inflammatory milieu of the degenerated disc. For successful regeneration of the IVD, the selection of a proper cell type and source is crucial. Wharton’s jelly-derived MSCs (WJ-MSCs) have gained significant interest and attention for clinical application because of their hypoimmunogenicity and immunomodulatory potential compared with MSCs from other sources[2-6]. WJ-MSCs exhibit very low expression of Human Leukocyte Antigen (HLA) class I and do not express class II (HLA-DR) as well as co-stimulatory antigens such as CD86, CD80, the key players to activate B cell and T cell[7, 8]. Furthermore, WJ-MSCs synthesize extensive amounts of tolerogenic IL-10, up-regulate TGF-β and express HLA-G[9]. Their low cellular immunogenecity is advantageous for allogeneic and xenogeneic transplantation. WJ-MSCs also possess immunomodulatory potential [6, 10]. Their immunomodulatory mechanism can be explained as : upregulation of negative co-stimulatory ligands; secretion of immunosuppressive soluble factors; generation of memory cells; cell fusion to escape recognition; immune avoidance mechanisms specific to fetal-maternal interface; attenuation of antigen-presenting cell functions; altered migration of immune cells, and T cell energy apoptosis tolerance[6, 10]. Thus, WJ-MSCs have been considered to be optimal candidates for cellular therapies in allogenic transplantation and used for treatment of various diseases (e.g., cancer, chronic liver disease, cardiovascular diseases, cartilage and tendon injury, immune mediated disorders, such graft versus host disease and systemic lupus erythematosus )[6, 10].
Also, the authors should appropriate define the cells that they intended to use in the current study based on the recent publication of MSCs commitee of ISCT (Viswanathan S, Shi Y, Galipeau J, Krampera M, Leblanc K, Martin I, Nolta J, Phinney DG, Sensebe L. Mesenchymal stem versus stromal cells: International Society for Cell & Gene Therapy (ISCT®) Mesenchymal Stromal Cell committee position statement on nomenclature. Cytotherapy 2019; 21: 1019-1024 [PMID: 31526643 DOI: 10.1016/j.jcyt.2019.08.002]). Should be Mesenchymal Stem Cells or Stromal cells.
Response:We greatly appreciate the reviewer’s concern. As reviewer suggested, we thoroughly investigated on this matter. Based on the following facts, we would like to rename our used cells as ‘wharton’s jelly- derived mesenchymal stromal cells (WJ-MSCs)’ instead of ‘wharton’s jelly derived stem cells’, which has been updated in the revised manuscript.
As per the latest latest statement released by The International Society for Cell & Gene Therapy (ISCT®), Mesenchymal Stromal Cell (ISCT MSC) committee offers a position statement to clarify the nomenclature of mesenchymal stromal cells (MSCs)[11]. The ISCT MSC committee continues to support the use of the acronym “MSCs” but recommends this be (i) supplemented by tissue-source origin of the cells, which would highlight tissue-specific properties; (ii) intended as MSCs unless rigorous evidence for stemness exists that can be supported by both in vitro and in vivo data; and (iii) associated with robust matrix of functional assays to demonstrate MSC properties, which are not generically defined but informed by the intended therapeutic mode of actions.
Similarly, the International Society for Cell & Gene Therapy (ISCT®) Mesenchymal Stromal Cell (ISCT MSC) committee issued a position paper in 2005 clarifying that the term mesenchymal stem cell is not equivalent or interchangeable with mesenchymal stromal cell (MSC). The former refers to a stem cell population with demonstrable progenitor cell functionality of self-renewal and differentiation whereas the latter refers to a bulk population with notable secretory, immunomodulatory and homing properties.
Clinical applications of WJ-MSCs can be attributed to five important biological properties: (a) homing to sites of inflammation following tissue injury when injected intravenously; (b) the secretion of multiple bioactive molecules capable of stimulating recovery of injured cells and inhibiting inflammation; (c) modulating the immune functions, (d) differentiation into various cell types, and finally (e) as a tool for gene therapy [12]. In addition to that, until 2004 when the first report providing robustevidence that WJ-stromal cells can be classified as MSCs waspublished[13].First discovered in the early seventies, MSCs, a populationof plastic-adherent, non-hematopoietic, fibroblast-like cells, werefirst isolated from bone marrow (BM). Afterward, in 1991, the term “mesenchymal stem cell”also known as “multipotent mesenchymal stromal cells,” wasproposed based on their properties.
Comments#2:5. Methods - 5.2 In vitro 3D hydrogel cell survival, lines 277-288. The authors should report the origin of WJ-MSCs. These cells were commercially available or provided by another laboratory. Due to Bioethics resrtictions, signed informed consent by the mothers should be reported in this section. Also, the informed consent should be according the declaration of Helsinki.
Response#2: We are thankful to the reviewer, WJ-MSCs were purchased from the Good Manufacturing Practice (GMP) facility of CHA Biotech (Seongnamsi, Korea) and this study was conducted according to the World Medical Association Declaration of Helsinki. With the written informed consent and ethics committee approval (No. BD2014-055), umbilical cord was obtained immediately after birth. This information has been updated in the revised manuscript (line 346-349)
Comments#3:5. Methods - 5.3 In vitro post-injection cell viability, lines 289-300
Quality control assessment of the WJ-MSCs should be performed by the authors and the results could be added either in the main manuscript or in supplementary section.
Response#3: We are thankful to the reviewer, quality control assessment of the WJ-MSCs has been updated in the revised manuscript (lines 117-129, 350-361, and 743-750) as given below.
Methods
WJ-MSCs were prepared as described elsewhere[3] and all culture conditions adhere to GMP standards. Briefly, the cord was washed with phosphate-buffered saline (PBS) and wharton’s jelly was then cut into pieces smaller than 5 mm3 after blood vessels were removed. The minced Wharton’s jelly was digested for 6–10 h in a sterilized bottle with 15 ml culture medium containing collagenase of type I at 0.075 % in 5 % carbon dioxide, at 37 °C with agitation in an incubator. The cells were then washed three times with D-Hank’s salt solution and centrifuged at 250 × g for 10 min at room temperature, and the cells were resuspended in culture medium DMEM with low glucose (DMEM-L; Gibco BRL) supplemented with 10 % (v/v) fetal bovine serum (FBS;Gibco BRL) and 1 % antibiotic–antimycotic solution (Gibco BRL) in humidified air with 5 % carbon dioxide at 37 °C. The cells were harvested after reaching 70–80% confluency and characterization tests were performed at passage 7. Quality control of these cells were performed according to the standards of the Korea’s Ministry of Food and Drug Safety.
Results:
2.1. Quality control of WJ-MSCs
The WJ-MSCs in the second, fourth, and seventh generations showed a spindle fibroblast-like shape (Suppl. Fig. 1A). After induction with adipogenic medium, WJ-MSCs gradually changed from fibroblast-like cells to flattened cells, and lipid droplets accumulated within them. The adipogenic differentiated MSCs were visualized by staining with Oil red-O on day 15. After incubation with osteogenic or chondrogenic medium for 15 days, MSCs were positive for alizarin red and alcian blue staining, respectively. (Suppl. Fig. 1B). Cell population doubling time was calculated three times. The cell doubling time in the second, fourth, and seventh generation were 22.35±0.53 h, 17.93±0.72 h, and 18.63±1.1 h, respectively, and the difference between the two groups was not significant(Suppl. Fig. 1C). Flow cytometry analysis of expressed surface antigens showed that these cells were uniformly positive for CD44, CD73, CD105, and CD90, and negative for the hematopoietic lineage markers CD45 (Suppl. Fig. 1D). No structural or numerical chromosomal abnormalities were found in karyotype analyses of the cells until P14.
Supplementary Figure 1. (A) Morphology of WJ-MSCs in the second, fourth, and seventh generation. P2, second generation; P4; fourth generation; P7; seventh generation (lens: ×40, ×100 and ×200 respectively). (B) Cellular multipotent differentiation of WJ-MSCs. Following adipogenic induction for 14 days, WJ-MSCs were positive for Oil red-O staining. Following osteogenic and chondrogenic induction for 14 days, WJ-MSCs were positive for alizarin red staining and alcian blue staining, respectively. Scale bars, 10 µm. (C) cell population doubling time. (D) flow cytometric analysis of cell surface markers of WJ-MSCs at P7. (E) chromosome karyotype analysis of the cultured WJ-MSCs.
Comments#4:5. Methods - 5.5 Experimental design in rat disc degenration model, lines 319 - 327. The authors mentioned that for their experiments they used WJ-MSCs of passage 7. In this passage some morphological features and characteristics may be changed. The evaluation of the quality characteristics of WJ-MSCs should also be extended in this passage.
Response#4: It has been reported that WJ-MSCs, even in the first six or seven passages, seem to possess good quantitative and qualitative traits (clonal ability, etc.)[14]. Our data showed that WJ-MSCs maintained high viability and pluripotency without any karyotype abnormalities until the nineteenth passage (Suppl. Fig. 1). The seventh passage of WJ-MSCs was therefore used for our study. In detail, quality characteristics have been explained in Response#3, line 117-129 and line 350-361 in the revised manuscript.
Comments#5:5. Methods - 5.7 RNA isolation and real time RT-PCR. The authors evaluated the expression of ADAMTS4, COX-2, MMP-13, and iNOs. However, further evaluation in proteomic level (e.g. Western Blot) should be performed, in order the results to be accurate. Also according to genes nomenclature, if the above genes are referred to rats should be written Adamts4, Cox-2 and etc.
Response#5:We greatly appreciate the reviewer’s concern. As the interrogation of biological systems reveals complex patterns of regulation between RNA and protein expression, co-analysis of these distinct cellular components become necessary. For example, although correlated expression of specific mRNA and protein is expected, post transcriptional regulation can lead to the decoupling of relationship. qPCR has several distinct advantages over other methods for assessing mRNA levels. First, this technology is sensitive enough to detect even a single copy of a gene sequence found within a sample. Second, qPCR is high throughput, allowing for the detection of a greater number of gene targets in less time as compared with, semi-quantitative methods. There is not a one to one correspondence between the amount of mRNA expression and subsequent production of protein.
In the earlier study, authors compared the results of PCR assay and Western blot analysis in a series of dyspeptic patients with unknown H. pylori status in order to establish the concordance of these two methods in the assessment of CagA status. PCR and Western blot analysis failed to provide comparable data in many cases. Western blot analysis seems to be more likely to give misleading results than PCR. Thus, PCR seems to be the method of choice to assess CagA status[15]. The main reason for this disagreement seems to be due to the methodological limits of Western blot analysis, which could be responsible for the false positive and negative results in the present series of patients, as also suggested by other authors.
Based on the above reasons we prioritized PCR analysis rather than western blot and other biochemical analysis. Herein, we have performed PCR analysis for five times (independently) taking five different samples from each group. Among them, results of three independent analysis showing promising results were used for statistical analysis. We appreciate the reviewer’s suggestions, due to very limited revision time we are unable to perform western blot this time, however, we are considering to perform western blot at the time of further investigation on this study.
We are thankful to the reviewer for raising this important point. In the revised manuscript, genes are written as per genes nomenclature, as suggested.
Comments#6 In addition, the authors should discuss the new results and try to compare their results with other related studies.
Response#6: we appreciate the reviewer’s suggestions we included the following part in discussion, in the revised manuscript (line 244-250 and line 309-312).
Our quality control assessment for WJ-MSCs demonstrated that WJ-MSCs showed a spindle fibroblast-like shape until fourteenth generations, which was consistent with our previous report[3]. Flow cytometric analysis of expressed surface antigens showed that these cells were uniformly positive for CD44, CD73, CD105, and CD90, and negative for the hematopoietic lineage markers CD45. Moreover, no structural or numerical chromosomal abnormalities were found in karyotype analyses of the cells until P14.
Hence, the regenerative effects of HAMC/WJ-MSCs in our study might therefore be linked with immunomodulatory and anti-inflammatory effects via paracrine signaling, regardless of whether WJ-MSCs differentiated into NP-like cells. In addition, HAMC provided a superior carrier for WJ-MSCs and improved regeneration in a rat model of disc degeneration.
Comments#73. Discussion, lines 247 - 248. Please remove the term first from the following sentence. To the best of our knowledge, this study is the first...HAMC hydrogel. Do the same in line 261.
Response#7:We are thankful to the reviewer. As suggested, we have removed the terminology ‘first’ and revised the sentences accordingly(line 313-314 and line 329-330).
We greatly appreciate the reviewers’ support in strengthening this manuscript, and we thank you for considering our manuscript for review. With the above-outlined revisions, we believe that our manuscript is considerably improved and now acceptable for publication in International Journal of Molecular Sciences. Please do not hesitate to contact me with additional comments or questions.
Thank you,
Sincerely,
In-Bo Han, M.D., PhD
Professor
Department of Neurosurgery, Spine Center
CHA University, CHA Bundang Medical Center
59, Yatap-ro, Bundang-gu, Seongnam-si, Gyeonggi-do, 13496, Korea
Tel: 031-780-1924, Fax: 031-780-5269
- Meisel, H.-J.; Agarwal, N.; Hsieh, P. C.; Skelly, A.; Park, J.-B.; Brodke, D.; Wang, J. C.; Yoon, S. T.; Buser, Z., Cell therapy for treatment of intervertebral disc degeneration: A systematic review. Global spine journal 2019, 9, (1_suppl), 39S-52S.
- Nekanti, U.; Rao, V. B.; Bahirvani, A. G.; Jan, M.; Totey, S.; Ta, M., Long-term expansion and pluripotent marker array analysis of Wharton’s jelly-derived mesenchymal stem cells. Stem cells and development 2010, 19, (1), 117-130.
- Ahn, J.; Park, E.-m.; Kim, B. J.; Kim, J.-S.; Choi, B.; Lee, S.-H.; Han, I., Transplantation of human Wharton’s jelly-derived mesenchymal stem cells highly expressing TGFβ receptors in a rabbit model of disc degeneration. Stem cell research & therapy 2015, 6, (1), 190.
- Liu, S.; Hou, K. D.; Yuan, M.; Peng, J.; Zhang, L.; Sui, X.; Zhao, B.; Xu, W.; Wang, A.; Lu, S., Characteristics of mesenchymal stem cells derived from Wharton's jelly of human umbilical cord and for fabrication of non-scaffold tissue-engineered cartilage. Journal of bioscience and bioengineering 2014, 117, (2), 229-235.
- Mallis, P.; Michalopoulos, E.; Chatzistamatiou, T.; Stavropoulos-Giokas, C., Mesenchymal stromal cells as potential immunomodulatory players in severe acute respiratory distress syndrome induced by SARS-CoV-2 infection. World Journal of Stem Cells 2020, 12, (8), 731-751.
- K Batsali, A.; Kastrinaki, M.-C.; A Papadaki, H.; Pontikoglou, C., Mesenchymal stem cells derived from Wharton's Jelly of the umbilical cord: biological properties and emerging clinical applications. Current stem cell research & therapy 2013, 8, (2), 144-155.
- Jyothi Prasanna, S.; Sowmya Jahnavi, V., Wharton's jelly mesenchymal stem cells as off-the-shelf cellular therapeutics: a closer look into their regenerative and immunomodulatory properties. The Open Tissue Engineering and Regenerative Medicine Journal 2011, 4, (1).
- Marino, L.; Castaldi, M. A.; Rosamilio, R.; Ragni, E.; Vitolo, R.; Fulgione, C.; Castaldi, S. G.; Serio, B.; Bianco, R.; Guida, M., Mesenchymal Stem Cells from the Wharton’s Jelly of the Human Umbilical Cord: Biological Properties and Therapeutic Potential. International Journal of Stem Cells 2019, 12, (2), 218.
- Prasanna, S. J.; Gopalakrishnan, D.; Shankar, S. R.; Vasandan, A. B., Pro-inflammatory cytokines, IFNγ and TNFα, influence immune properties of human bone marrow and Wharton jelly mesenchymal stem cells differentially. PloS one 2010, 5, (2), e9016.
- Kim, D.-W.; Staples, M.; Shinozuka, K.; Pantcheva, P.; Kang, S.-D.; Borlongan, C. V., Wharton’s jelly-derived mesenchymal stem cells: phenotypic characterization and optimizing their therapeutic potential for clinical applications. International journal of molecular sciences 2013, 14, (6), 11692-11712.
- Viswanathan, S.; Shi, Y.; Galipeau, J.; Krampera, M.; Leblanc, K.; Martin, I.; Nolta, J.; Phinney, D.; Sensebe, L., Mesenchymal stem versus stromal cells: International Society for Cell & Gene Therapy (ISCT®) Mesenchymal Stromal Cell committee position statement on nomenclature. Cytotherapy 2019, 21, (10), 1019-1024.
- Kamal, M. M.; Kassem, D. H., Therapeutic potential of wharton’s jelly mesenchymal stem cells for diabetes: achievements and challenges. Frontiers in Cell and Developmental Biology 2020, 8.
- Wang, H. S.; Hung, S. C.; Peng, S. T.; Huang, C. C.; Wei, H. M.; Guo, Y. J.; Fu, Y. S.; Lai, M. C.; Chen, C. C., Mesenchymal stem cells in the Wharton's jelly of the human umbilical cord. Stem cells 2004, 22, (7), 1330-1337.
- Christodoulou, I.; Kolisis, F.; Papaevangeliou, D.; Zoumpourlis, V., Comparative evaluation of human mesenchymal stem cells of fetal (Wharton's jelly) and adult (adipose tissue) origin during prolonged in vitro expansion: considerations for cytotherapy. Stem cells international 2013, 2013.
- Paoluzi, O. A.; Rossi, P.; Montesano, C.; Bernardi, S.; Carnieri, E.; Marchione, O. P.; Nardi, F.; Iacopini, F.; Pica, R.; Paoluzi, P., Discrepancy between polymerase chain reaction assay and Western blot analysis in the assessment of CagA status in dyspeptic patients. Helicobacter 2001, 6, (2), 130-135.
Reviewer 2 Report
The authors combined a hydrogel consisting of hyaluronan and methylcellulose with Wharton’s Jelly-derived mesenchymal stem cells to inject it into degenerated intervertebral discs for disc regeneration in the rat model. They found that this combination led to better outcome compared to the control groups included in the study (only biomaterial and only cells). They investigated cartilage ECM markers and inflammation and degradation-associated factors such as MMP13, ADAMTS, iNOS, COX-2. They did not track the implanted cells - since it would be interesting to prove their survival since the authors checked the effect of shear forces on vitality in vitro and found rather low cell vitality rates. Otherwise trophic effects of the cells should be discussed more thoroughly.
They did not comment on the degradability of the biomaterial, particularly the MC component.
line 60: it might be rather osteoarthritis than arthritis
line 77: it requires an explanation how MSCs differentiate into osteoclasts (which type of stem cells? Were blood derived MSCs from the monocyte lineage used in this study?) and the next question is, how do osteoclasts produce osteophytes?
How about the cytotoxicity of the biomaterial, since 52% cell survival is low (line 104). Is the high death rate only caused by shear forces? TissueFill is already commercially available for the clinics. how is the low cell survival rate explanable? since it is caused by shear forces how could these forces be avoided - other application strategies. A comparison to cell survival with no injection simply after embedding in the gel would be interesting.
line 142: which histological score was used.
line 163 and line 200: the brackets are confusing.
line 221 and line 241-242: cell survival after in vivo injection: was an anti-human antibody used to check the fate of the implanted cells? which procedure was undertaken to track the implanted cells?
line 224: insert a blank before the bracket
line 239: "correct phenotype" what means correct? Please explain.
line 270: physical blend: please explain it, what does it mean? Is it cross-linked? How about its long-term stability/degradation?
line 275: Later it is always abbreviated
line 274: "Tissue Fill" why was it selected for in vitro but not in vivo analyses as a reference?
line 282: Calcium AM - please explain this abbreviation
line 287: there seems to be a surplus blank
line 293: was it dissolved in growth medium?
Line 305: how long were animals acclimatized before the experiment starts?
320: Fifty rats were randomly divided into four groups: (n=10 per group). What happened with the remaining 10 animals. Please state how many animals were included in each method and how many independent experiments were performed e.g. in regard to PCR etc. In the figure legends there is stated n=3 – does it mean independent experiments?
line 321: "the uninjured..." please revise this sentence.
line 326: something is wrong with the brackets. What do the arrows mean?
line 339: MRI index, please provide a reference, also for the histpathological score (lines 369-370).
line 374: "humanely" should be omitted because it remains a matter of debate…
Line 380: do not write Aggrecan and Collagen using capital letters, compare page 4
Figure 1: why does the viability decrease so much after 7 days in the hydrogels?
Figure 2: an optical density „OD“ is shown. How many cells really survive post injection?
Figure 3: was the Safranin O staining quantified?
Author Response
Dear Ms. Nikolina Popovic,
We would like to thank you and the reviewers for considering our manuscript “An injectable hyaluronan-methylcellulose (HAMC) hydrogel combined with Wharton’s jelly-derived mesenchymal stem cells (WJ-MSCs) promotes degenerative disc repair” (ijms-939826) and providing us the valuable feedback. We have revised our manuscript according to the reviewer’s comments as detailed below. Moreover, a list of changes is highlighted in red in the revised manuscript.
Response to Reviewer:
Comments#1:The authors combined a hydrogel consisting of hyaluronan and methylcellulose with Wharton’s Jelly-derived mesenchymal stem cells to inject it into degenerated intervertebral discs for disc regeneration in the rat model. They found that this combination led to better outcome compared to the control groups included in the study (only biomaterial and only cells). They investigated cartilage ECM markers and inflammation and degradation-associated factors such as MMP13, ADAMTS, iNOS, COX-2. They did not track the implanted cells - since it would be interesting to prove their survival since the authors checked the effect of shear forces on vitality in vitro and found rather low cell vitality rates. Otherwise trophic effects of the cells should be discussed more thoroughly.
Response#1: We are thankful to the reviewer, as suggested we have given the explannation as follows, and updated the revised manuscript accordingly (line 288-297)
In the present study, however, we did not track the implanted WJ-MSCS and could not confirm survival of implanted WJ-MSCS at 6 weeks post implantation. In terms of survival of implanted cells, there have been several reports on long-term survival (more than 3 months) of implanted MSCs loaded into biomaterials [1, 2]. In our previous study, WJ-MSCs-loaded TissueFill (1% cross-linked hyaluronic acid hydrogel) was intradiscally implanted into degenerated rabbit disc and no survived cell were found 12 weeks post implantation [3]. Here, we confirmed that a complex mixture of cytokines produced by WJ-MSCs including transforming growth factor beta (TGF-β) ligands (TGFβ1, TGFβ2, and TGFβ3), growth differentiation factor-15 (GDF-15), chemokine (C-C motif) ligand 5 (CCL5), and MMP1 could trigger multiple signaling system including TGF-β signaling and stimulate IVD regeneration.
They did not comment on the degradability of the biomaterial, particularly the MC component.
Response#1: We are thankful to the reviewer, as suggested, we have explained it the revised manuscript (line258-269), as below:
The degradability of the HAMC was not tested in this study. However, the Shoichet lab has previously tested the degradation rate of HAMC when implanted in the intrathecal space of the spinal cord or the subretinal space of the retina, reporting that HA and MC are largely absent in approximately 3 and 7 days respectively, HA through enzymatic degradation by hyaluronidases, and MC from bulk resorption (Ballios et al., 2010; Austin et al., 2012). We have updated the manuscript to include this information:
Line 102: HAMC spatially localizes the drug or cells of interest at the site of delivery and promotes short-term controlled release of drugs to the central nervous system (CNS), with a degradation time of approximately 3 to 7 days in vivo [20, 21]
Ballios, B.G., et al., A hydrogel-based stem cell delivery system to treat retinaldegenerative diseases. Biomaterials, 2010. 31(9): p. 2555-2564.
Austin, J.W., et al., The effects of intrathecal injection of a hyaluronan-based hydrogelon inflammation, scarring and neurobehavioural outcomes in a rat model of severespinal cord injury associated with arachnoiditis. Biomaterials, 2012. 33(18): p. 4555-4564.
line 60: it might be rather osteoarthritis than arthritis
Response: We are thankful to the reviewer. Its has been corrected in the revised manuscript (Line 60).
line 77: it requires an explanation how MSCs differentiate into osteoclasts (which type of stem cells? Were blood derived MSCs from the monocyte lineage used in this study?) and the next question is, how do osteoclasts produce osteophytes?
Response: We greatly appreciate the reviewer’s concern. also, we apologize for the misleading claim. Based on the following evidences we have revised our sentences and updated in the revised manuscript (line 93-96)
In earlier study Vadala G. et. al., 2012 investigated that IVD degeneration in rabbits and investigated the osteophyte growth in the anterolateral disc space due to cell leakage after MSC transplantation. Additionally, histological analysis showed that the osteophytes were composed of mineralized tissue surrounded by chondrocytes, with the labelled MSCs among the osteophyte-forming cells. The labelled MSCs were not found in the nucleus. Inflammatory cells were not observed in any injected IVDs. This further raised the concern that MSCs can migrate out of the nucleus and undesirable bone formation may occur[4]. In that study, labelled MSCs have been found among osteophyte-forming cells suggesting a contributory role in the osteophyte formation process through the endochondral ossification process.
The growing concern is, osteophytes, as can be found at joint margins, totally derived from chondrocyte precursors or is there an attribution of other cells such as (pre)osteoblast? In subsequent study, in animal models of osteoarthritis (OA) it was observed that in addition to chondrogenesis and endochondral ossification intramembraneous lamellar bone formation contributes to the definitive osteophyte. Developing osteophytes were composed of fibroblasts, mesenchymal prechondrocytes, maturing chondrocytes, hypertrophic chondrocytes and osteoblast. Osteophyte formation closely resembles the process of chondrogenesis and endochondral bone formation as can be seen during embryogenesis [5].
How about the cytotoxicity of the biomaterial, since 52% cell survival is low (line 104). Is the high death rate only caused by shear forces? TissueFill is already commercially available for the clinics. how is the low cell survival rate explanable? since it is caused by shear forces how could these forces be avoided - other application strategies. A comparison to cell survival with no injection simply after embedding in the gel would be interesting.
Response We thank the reviewer for their comments. It is well established in the literature that shear forces can damage cell membranes and induce cell death when injected through a fine-gauge needle (Aguado et al., 2012; Connolly et al., 2020; Rossetti et al., 2016; Wahlberg et al., 2018). We have included these references in the manuscript:
Rossetti, T., Nicholls, F. and Modo, M., 2016. Intracerebral cell implantation: preparation and characterization of cell suspensions. Cell transplantation, 25(4), pp.645-664.
Aguado, B.A., Mulyasasmita, W., Su, J., Lampe, K.J. and Heilshorn, S.C., 2012. Improving viability of stem cells during syringe needle flow through the design of hydrogel cell carriers. Tissue Engineering Part A, 18(7-8), pp.806-815.
Connolly, S., McGourty, K. and Newport, D., 2020. The in vitro inertial positions and viability of cells in suspension under different in vivo flow conditions. Scientific Reports, 10(1), pp.1-13.
Wahlberg, B., Ghuman, H., Liu, J.R. and Modo, M., 2018. Ex vivo biomechanical characterization of syringe-needle ejections for intracerebral cell delivery. Scientific reports, 8(1), pp.1-17.
Considering that cell viability was measured directly following injection through a needle, it is unlikely that the decrease in viability is from other effects that would take more time to manifest. One other potential contributor to the decrease in cell viability is the process of dissociating the WJ-MSCs from the dish, which was not measured separately. We have previously shown for human induced pluripotent stem cell-derived neuroepithelial cells that removal from the culture dish can also induce anoikis and cell death (Payne et al., 2019). Although it is beyond the scope of this project, future studies could look at reducing the injection rate or using a different concentration of HAMC to test the effect on cell viability and whether it can be improved. We can make the comparison of injection in HAMC vs mixing in HAMC if you compare Figures 1 and 2, where we see that after one day of culture in 3D HAMC WJ-MSC viability remains at ~90% whereas following injection it is reduced almost by half, which suggests that the HAMC itself is not cytotoxic but rather it is the shear stress experienced by cells that affects their survival.
Payne, S.L., Tuladhar, A., Obermeyer, J.M., Varga, B.V., Teal, C.J., Morshead, C.M., Nagy, A. and Shoichet, M.S., 2019. Initial cell maturity changes following transplantation in a hyaluronan-based hydrogel and impacts therapeutic success in the stroke-injured rodent brain. Biomaterials, 192, pp.309-322.
line 142: which histological score was used.
Response: We thank the reviewer, we have written the method (5.8 Safranin-O staining) for the Histological score and described in detail in line 447-460.
Briefly, for the assessment of histologic structure, a 14-point score was used based on safranin-O staining [6]. At 6 weeks post-implantation, the scoring consists of five parameters: NP structure, NP clefts/fissures, annulus fibrosus (AF)/NP boundary, AF structure, and AF/clefts/fissures. The sum of the separate scores ranges from 0 (normal) to 14 (most severe)[6].
line 163 and line 200: the brackets are confusing.
Response:Thank you,corrected in the revised manuscript (line 194-195 and line 231 ).
Line 221 and line 241-242: cell survival after in vivo injection: was an anti-human antibody used to check the fate of the implanted cells? Which procedure was undertaken to track the implanted cells?
Response: In the present study, however, we did not track the implanted WJ-MSCS and could not confirm survival of implanted WJ-MSCS at 6 weeks post implantation. In terms of survival of implanted cells, there have been several reports on long-term survival (more than 3 months) of implanted MSCs loaded into biomaterials [1, 2].
line 224: insert a blank before the bracket
Response: Thank you,corrected in the revised manuscript (line 272-273).
line 239: "correct phenotype" what means correct? Please explain.
Response:we appreciates the reviewer’s concern. Herein, correct phenotype means Nucleus pulposus like(NP) phenotype. Which has been updated in the revised manuscript (line 288).
line 270: physical blend: please explain it, what does it mean? Is it cross-linked? How about its long-term stability/degradation?
Response: The degradability of the HAMC was not tested in this study. However, the Shoichet lab has previously tested the degradation rate of HAMC when implanted in the intrathecal space of the spinal cord or the subretinal space of the retina, reporting that HA and MC are largely absent in approximately 3 and 7 days respectively, HA through enzymatic degradation by hyaluronidases, and MC from bulk resorption (Ballios et al., 2010; Austin et al., 2012). We have updated the manuscript to include this information:
Line 102: HAMC spatially localizes the drug or cells of interest at the site of delivery and promotes short-term controlled release of drugs to the central nervous system (CNS), with a degradation time of approximately 3 to 7 days in vivo [20, 21]
Ballios, B.G., et al., A hydrogel-based stem cell delivery system to treat retinaldegenerative diseases. Biomaterials, 2010. 31(9): p. 2555-2564.
Austin, J.W., et al., The effects of intrathecal injection of a hyaluronan-based hydrogelon inflammation, scarring and neurobehavioural outcomes in a rat model of severespinal cord injury associated with arachnoiditis. Biomaterials, 2012. 33(18): p. 4555-4564.
line 275: Later it is always abbreviated
Response:Thank you,corrected in the revised manuscript (line 343).
line 274: "Tissue Fill" why was it selected for in vitro but not in vivo analyses as a reference?
Response: We thank the reviewer, herein we have explored the suitability of HAMC as a potential cell delivery vehicle, and Tissue Fill was used as a reference scaffold in in-vitro. Our result showed that HAMC was found to be superior delivery vehicle compared to tissuefill. Therefore, therefore we didn’t tested its effect in in-vivo, additionally, we have already examined the effect of tissuefill as delivery vehicle in our earlier studies.
line 282: Calcium AM - please explain this abbreviation
Response: We apologize for the mislead caused to the reviewer, that is ‘calcein AM’ instead of ‘calcium AM’. Which has been corrected in the updated manuscript (line 367). Calcein AM(AM = acetoxymethyl) is a cell-permeable dye that can be used to measure cell viability in most eukaryotic cells. Live cells are distinguished by the presence of ubiquitous intracellular esterase activity, determined by the enzymatic conversion of the virtually nonfluorescent cell-permeant calcein AM to the intensely fluorescent calcein. The polyanionic dye calcein is well retained within live cells, producing an intense uniform green fluorescence in live cells (ex/em ~495 nm/~515 nm).
line 287: there seems to be a surplus blank
Response: Thank you,corrected in the revised manuscript (line 372).
line 293: was it dissolved in growth medium?
Response: The HAMC was dissolved in PBS. The manuscript has been updated to include that information: Line 350-361 and line 378: WJ-MSCs were trypsinized and 2 x 104 cells/uL were suspended into equal volume of 1% HAMC, final concentration 0.5% HA/ 0.5% MC in PBS.
Line 305: how long were animals acclimatized before the experiment starts?
Response:Animals were acclimatized for one week, which has been updated in the revised manuscript (line 390-391).
320: Fifty rats were randomly divided into four groups: (n=10 per group). What happened with the remaining 10 animals. Please state how many animals were included in each method and how many independent experiments were performed e.g. in regard to PCR etc. In the figure legends there is stated n=3 – does it mean independent experiments?
Response: We are thankful to the reviewer, and we apologize for the mistake, total numbers of animal used in this study were ‘forty’ instead of ‘fifty’. This has been updated in the revised manuscript. Numbers of animals used for each experiment are mentioned in the figure legends of respective experimental results, in the manuscript. All the experiments were performed by taking different/ separate animals (independently). So, n=3 means three independent experiments in this study.
line 321: "the uninjured..." please revise this sentence.
Response: Thank you,corrected in the revised manuscript (line 409).
line 326: something is wrong with the brackets. What do the arrows mean?
Response: Thank you,corrected in the revised manuscript (line412-440).
line 339: MRI index, please provide a reference, also for the histpathological score (lines 369-370). Response: we are thankful to the reviewer, references have been provided at the respective places in the revised manuscript (line 427 and 458).
line 374: "humanely" should be omitted because it remains a matter of debate…
Response: Thank you,corrected in the revised manuscript (line 462).
Line 380: do not write Aggrecan and Collagen using capital letters, compare page 4
Response: Thank you,corrected in the revised manuscript (line 463).
Figure 1: why does the viability decrease so much after 7 days in the hydrogels?
Response: Although we did not test the exact mechanism by which cell death is occurring, it is likely due to a combination of factors. The cells were cultured in the hydrogels in suspension with no additional media or supplements added for 7 days and compared to standard 2D culture. It is likely that the lack of media addition contributed to low cell viability by day 7. Furthermore, HAMC is reported to be approx. 90% degraded in vitro at 14 days (Gupta et al., 2006), and therefore it is likely that by day 7 the mechanical strength of HAMC and TissueFill is reduced which may lead to reduced cell support. Lastly, although HA has been previously reported to promote cell adhesion and survival through CD44 signaling (Ballios et al., 2015) it is unknown if the WJ-MSCs used here express CD44, which may also contribute to the reduction of cell viability over time.
We have included a discussion of this in the main manuscript:
Line 258:
We also saw a decrease in WJ-MSC viability after 7 days of culture in HAMC or TissueFill when compared with 2D standard culture, although the viability in HAMC was significantly greater than in TissueFill. The cells were cultured in the hydrogels in suspension with no additional media or supplements in order to challenge the cells with stress, which may have contributed to their decrease in viability over time. Furthermore, HAMC is reported to be approx. 90% degraded in vitro at 14 days (Gupta et al., 2006), and therefore it is likely that by day 7 the mechanical strength of HAMC and TissueFill is reduced which may lead to reduced cell support.
Gupta, D., Tator, C.H. and Shoichet, M.S., 2006. Fast-gelling injectable blend of hyaluronan and methylcellulose for intrathecal, localized delivery to the injured spinal cord. Biomaterials, 27(11), pp.2370-2379.
Ballios, B.G., Cooke, M.J., Donaldson, L., Coles, B.L., Morshead, C.M., van der Kooy, D. and Shoichet, M.S., 2015. A hyaluronan-based injectable hydrogel improves the survival and integration of stem cell progeny following transplantation. Stem cell reports, 4(6), pp.1031-1045.
Figure 2: an optical density „OD“ is shown. How many cells really survive post injection?
Response: The CCK-8 assay uses the tetrazolium salt WST-8[2-(2-methoxy-4-nitrophenyl)-3-(4-nitrophenyl)-5-(2,4-disulfophenyl)-2H-tetrazolium,monosodium salt] which is reduced by dehydrogenases in cells to give a yellow colored product (formazan) which is soluble in tissue culture medium. Due to the nature of the CCK-8 assay, the readout is colorimetric change and therefore total cell number was not obtained.
Figure 3: was the Safranin O staining quantified?
Response: We performed histological score analysis through safranin-O staining because it reflects more closely with the extent of intervertebral disc damage.
We greatly appreciate the reviewers’ support in strengthening this manuscript, and we thank you for considering our manuscript for review. With the above-outlined revisions, we believe that our manuscript is considerably improved and now acceptable for publication in International Journal of Molecular Sciences. Please do not hesitate to contact me with additional comments or questions.
Thank you,
Sincerely,
In-Bo Han, M.D., PhD
Professor
Department of Neurosurgery, Spine Center
CHA University, CHA Bundang Medical Center
59, Yatap-ro, Bundang-gu, Seongnam-si, Gyeonggi-do, 13496, Korea
Tel: 031-780-1924, Fax: 031-780-5269
- Sobajima, S.; Vadala, G.; Shimer, A.; Kim, J. S.; Gilbertson, L. G.; Kang, J. D., Feasibility of a stem cell therapy for intervertebral disc degeneration. The spine journal 2008, 8, (6), 888-896.
- Sakai, D.; Mochida, J.; Yamamoto, Y.; Nomura, T.; Okuma, M.; Nishimura, K.; Nakai, T.; Ando, K.; Hotta, T., Transplantation of mesenchymal stem cells embedded in Atelocollagen® gel to the intervertebral disc: a potential therapeutic model for disc degeneration. Biomaterials 2003, 24, (20), 3531-3541.
- Ahn, J.; Park, E.-m.; Kim, B. J.; Kim, J.-S.; Choi, B.; Lee, S.-H.; Han, I., Transplantation of human Wharton’s jelly-derived mesenchymal stem cells highly expressing TGFβ receptors in a rabbit model of disc degeneration. Stem cell research & therapy 2015, 6, (1), 190.
- Vadalà, G.; Sowa, G.; Hubert, M.; Gilbertson, L. G.; Denaro, V.; Kang, J. D., Mesenchymal stem cells injection in degenerated intervertebral disc: cell leakage may induce osteophyte formation. Journal of tissue engineering and regenerative medicine 2012, 6, (5), 348-355.
- van der Kraan, P. M.; van den Berg, W. B., Osteophytes: relevance and biology. Osteoarthritis and cartilage 2007, 15, (3), 237-244.
- Tam, V.; Chan, W. C. W.; Leung, V. Y. L.; Cheah, K. S. E.; Cheung, K. M. C.; Sakai, D.; McCann, M. R.; Bedore, J.; Séguin, C. A.; Chan, D., Histological and reference system for the analysis of mouse intervertebral disc. Journal of Orthopaedic Research 2018, 36, (1), 233-243.
Round 2
Reviewer 1 Report
Dear Authors,
Thak you for revising your manuscript according to my suggestions.
Good work!
Reviewer 2 Report
My previous comments have been fully adressed and the manuscript was revised and improved accordingly. Hence, I have no major concerns.
There is only a very minor Problem which can be addressed during proof reading:
Lines 16, 44 and 76-79 etc.: insert blanks (analysis.We), (mesenchymalstromal), check all the novel sequences (disc.For….VD,the...) for lacking blanks
the whole novel sequences
line 124: remove surplus point
line 236: please correct: "high viscoisity"